# Inhibition of the STAT3/Fanconi anemia axis is synthetic lethal with PARP inhibition in breast cancer

Celia D. Rouault [1], Lucile Bansard[2], Elena Martínez-Balsalobre [3], Caroline Bonnet[1], Julien Wicinski [1], Shuheng Lin[1], Ludovic Colombeau [4], Sylvain Debieu[4], Guillaume Pinna [5], Marie Vandamme[5], Margot Machu[2], Olivier Rosnet [1], Véronique Chevrier[1], Cornel Popovici [6], Hagay Sobol[6], Rémy Castellano[7], Eddy Pasquier[8], Geraldine Guasch [1], Raphaël Rodriguez [4], Julie Pannequin[2], Jean-Marc Pascussi[2], Christophe Lachaud [3], Emmanuelle Charafe-Jauffret[1] ✉ & Christophe Ginestier [1] ✉

The targeting of cancer stem cells (CSCs) has proven to be an effective approach for limiting tumor progression, thus necessitating the identification of new drugs with anti-CSC activity. Through a high-throughput drug repositioning screen, we identify the antibiotic Nifuroxazide (NIF) as a potent anti-CSC compound. Utilizing a click chemistry strategy, we demonstrate that NIF is a prodrug that is specifically bioactivated in breast CSCs. Mechanistically, NIF-induced CSC death is a result of a synergistic action that combines the generation of DNA interstrand crosslinks with the inhibition of the Fanconi anemia (FA) pathway activity. NIF treatment mimics FA-deficiency through the inhibition of STAT3, which we identify as a non-canonical transcription factor of FA-related genes. NIF induces a chemical HRDness (Homologous Recombination Deficiency) in CSCs that (re)sensitizes breast cancers with innate or acquired resistance to PARP inhibitor (PARPi) in patient-derived xenograft models. Our results suggest that NIF may be useful in combination with PARPi for the treatment of breast tumors, regardless of their HRD status.

Two decades ago, functionally defined cancer stem cells (CSCs) were reliably identified in numerous solid tumors, including breast cancer[1], colorectal cancer[2], or glioblastoma[3], and gave first insight into intratumoral heterogeneity (ITH). CSC burden has been repeatedly associated with tumor progression and therapeutic failure[4–6]. Through recent advances in single-cell sequencing, the portrait of the ITH has been further revealed[7]. Far from the resultant of a rigid hierarchy initially described[8], the ITH reflects the co-existence of multiple cellular states driven by genetic, epigenetic, and microenvironmental influences. The transition between cellular states, the so-called cell plasticity, leads to tumor fitness, yielding cells with molecular programs that facilitate treatment failure and recurrence. While the

[1]CRCM, Inserm, CNRS, Institut Paoli-Calmettes, Aix-Marseille University, Epithelial Stem Cells and Cancer Lab, Equipe Labellisée LIGUE Contre Le Cancer, Marseille, France. [2]IGF, University Montpellier, CNRS INSERM, Montpellier, France. [3]CRCM, Inserm, CNRS, Institut Paoli-Calmettes, Aix-Marseille University, DNA Interstrand Crosslink Lesions and Blood Disorder Team, Marseille, France. [4]Institut Curie, CNRS, INSERM, Biomedicine Laboratory PSL Research University, Paris, France. [5]Plateforme ARN Interférence (PARI), Université Paris Cité, Inserm, CEA Stabilité Génétique Cellules Souches et Radiations, Fontenay-aux-Roses, France. [6]Aix-Marseille University, Cancer Genetic Laboratory, Cancer Biology Department Institut Paoli-Calmettes, Marseille, France. [7]CRCM, Aix-Marseille University, INSERM, CNRS, Institut Paoli-Calmettes, TrGET Plateform, Marseille, France. [8]CRCM, INSERM, CNRS, Institut Paoli-Calmettes, Aix-Marseille University, Reverse Molecular Pharmacology in Pediatric Oncology, Marseille, France. ✉e-mail: charafee@ipc.unicancer.fr; christophe.ginestier@inserm.fr

drivers of cellular plasticity remain poorly understood, consensus meta-programs defining cellular states in different tumors start to be identified[7,9]. Among them, EMT-like (Epithelial-to-Mesenchymal Transition) meta-programs define cells with a more plastic adaptive state and match with the actual CSC molecular definition. In this context, targeting CSC-state remains a prerequisite to overcoming therapeutic resistance. A significant number of studies have identified various regulatory networks that define CSC-state and their potential vulnerabilities[10]. Anti-CSC therapies have been tested in preclinical studies demonstrating that a reduction in the proportion of CSCs leads to a better response to conventional treatment, with a limitation of tumor progression and a reduction in metastatic spread. In breast cancer, salinomycin, and later its derivative ironomycin, was one of the first anti-CSC compounds identified, able to target the iron homeostasis dependency of CSC[11,12]. Other approaches appeared to sensitize CSC to chemotherapy or oncogene-targeted therapies[13–16]. Recently, it has been reported that pharmacological targeting of the EMT program following treatment with netrin-1-blocking antibody induces an effective reduction in tumor progression, alleviating resistance to standard treatments[17,18]. All these observations have prompted pharmaceutical companies to launch programs to test anti-CSC therapies in patients[10]. Although targeting CSC state is promising, very few anti-CSC therapies have reached phase III clinical trials. Among the hurdles that explain this relative failure, the strict rules that govern clinical trials have massively reduced the approval rate for oncology drugs over the last 40 years[19]. Indeed, in early stage of clinical trials, most anticancer compounds are selected on the basis of their ability to reduce tumor mass (RECIST criteria)[20]. Hence, this criterion selects drugs that have preferential impact on tumor bulk but not on the CSC state. Furthermore, anti-CSC compounds developed by pharmaceuticals companies are mainly focused on the targeting of core stemness programs such as WNT, NOTCH, or Hedgehog pathways[21]. However, these therapeutic strategies have since been shown to have a limited therapeutic window due to adverse side effects on the pool of normal stem cells that shares common regulatory pathways with CSC.

In this context, drug repurposing—i.e., the process of identifying new therapeutic uses for drugs already approved for specific indications—appears as an effective and attractive alternative for increasing the approval rate of anti-CSC therapies. Re-testing these compounds on their anti-CSC activity could shed light on drugs whose anti-tumor properties are yet to be demonstrated.

In this study, we take advantage of the high-content analysis strategy we have previously developed[22,23] to screen a library of FDA-approved compounds with the CSC fate as a readout. We identify an antibiotic, the Nifuroxazide (NIF), as a promising anti-CSC compound. We then demonstrate that NIF is a pro-drug selectively activated by ALDH1A1 in CSC and acts as a dual-action drug. First, it induces DNA interstrand crosslinks (ICLs) and then inhibits the Fanconi anemia/homologous recombination machinery required for ICL repair. Mechanistically, we show that NIF acts through the inhibition of STAT3, which we identify as a non-canonical transcription factor for Fanconi anemia genes. Finally, inspired by the therapeutic approach used to treat homologous recombination-deficient (HRD) breast, ovarian, and prostate cancer patients[24,25], we demonstrate that NIF treatment induces selective chemical HRDness in breast CSCs that is synthetically lethal with PARP inhibitor (PARPi). This strategy opens therapeutic perspectives in cancer treatment, with potential applications in patients with tumors that are HR-proficient or -deficient and resistant to PARPi.

## Results

### Drug repurposing screening identifies Nifuroxazide (NIF) as an anti-CSC compound

To explore whether we can requalify drugs as new anti-CSC therapies, we screened the 1280-compound PRESTWICK repurposing library at final concentration of 10 μM in four cancer cell lines (SUM159, SW620, MKN45, PANC1) issued from four different tumor types (breast, colon, gastric, and pancreatic cancer) (Fig. 1A). Each compound was systematically tested as separate triplicate in an ALDEFLUOR-probed CSC detection assay[26]. Of note, for all the models used, cells harboring a high ALDH enzymatic activity (ALDHbr) have all been functionally demonstrated to be enriched in CSC compared to the cell population presenting a low ALDH activity (ALDHneg)[27–29]. Screening data were analyzed by calculating the average CSC proportion upon compound treatment, relative to the vehicle (DMSO) used as control. Following data correction, hits were identified by a statistical Z-score integrating measurements from all four cancer cell lines. Our analysis revealed only one hit, for which drug treatment significantly decreased the CSC population (Z-score < −3.0), corresponding to the antibiotic Nifuroxazide (NIF) (Fig. 1B, C). Of note, other compounds with the same chemical structure (5-nitrofuran derivatives) were top-ranked in the list (Furaltadone hydrochloride, z-score < −1.9; Furazolidone, z-score < −1.5) suggesting a robust anti-CSC activity for this class of antibiotics (Fig. 1C). To further characterize the effect of NIF on the CSC population we first confirmed a reduction of the ALDHbr cell population in five additional cancer cell lines (Supplementary Fig. 1A). Then, we functionally validated this effect using a tumorsphere assay, reflecting CSC self-renewal in vitro, and demonstrated a constant reduction of the tumorsphere-forming efficiency in all NIF-treated cell lines (Fig. 1D and Supplementary Fig. 1B). Of note, NIF has been identified as a potent anti-multiple myeloma drug by screening the PRESTWICK repurposing library[30], thereby presenting an opportunity to elucidate underlying anti-cancer mechanisms. In this previous study, NIF was described as a potent STAT3 inhibitor. Therefore, we hypothesized that the anti-CSC effect of NIF might be mediated through this actionable target. We first confirmed that NIF treatment efficiently decreased STAT3 activation as demonstrated by the reduction of the ratio pSTAT3/STAT3 and of its transcriptional target Cyclin D1, in both cell subpopulations (ALDHneg and ALDHbr) (Fig. 1E and Supplementary Fig. 1C). Moreover, we performed a gene set enrichment analysis (GSEA) on RNA-seq data generated from ALDHbr SUM159 (breast cancer) and SW620 (colon cancer) cells under NIF treatment or in control condition (DMSO). We showed that NIF-treated ALDHbr cells were negatively associated with genes related to STAT3 pathway compared with the control, further confirming the potential of NIF to inhibit STAT3 activity (Supplementary Fig. 1D). To assess the effect of STAT3 inhibition on the CSC population, we inhibited STAT3 transcription using the CRISPRi dCas9-KRAB fusion protein in different cell lines (SW620-KRAB; SUM159-KRAB; S68-KRAB) (Supplementary Fig. 1E)[30]. Surprisingly, while in the colon cancer cell line (SW620-KRAB) the inhibition of STAT3 mimicked the effect of NIF treatment with a significant reduction of the ALDHbr cells proportion and tumorsphere-forming efficiency (Supplementary Fig. 1F, G), no effect was observed on the breast CSC population following STAT3 invalidation (Supplementary Fig. 1H, I). Similar results were obtained in the presence of compounds inhibiting STAT3 pathway (STAT3i, napabucasin and JAK2i, Ruxolitinib) (Supplementary Fig. 1J–N). Overall, these observations suggest that, in breast cancers, STAT3 inhibition is not sufficient to explain NIF effect on CSC. A possible explanation for NIF anti-CSC activity in breast cancer cells may reside in ALDH1's ability to bio-activate certain 5-nitrofuran derivatives. This metabolic activity leads to an oxidation and an inactivation of ALDH1 with the subsequent generation of 5-nitofuran metabolites[31,32]. Indeed, using colony-formation assay, we demonstrated that NIF treatment was significantly more toxic in the ALDHbr cell population than in ALDHneg cells (Fig. 1F and Supplementary Fig. 1O). We next inhibited ALDH enzymatic activity by stably transducing SUM159-KRAB and S68-KRAB with a sgALDH1A1 or a sgEmpty as a control (Supplementary Fig. 1P, Q). Interestingly, ALDH1A1 depletion by itself had no effect on the tumorsphere-forming efficiency of breast cancer cells, whereas it was sufficient to abrogate the capacity of NIF treatment to reduce

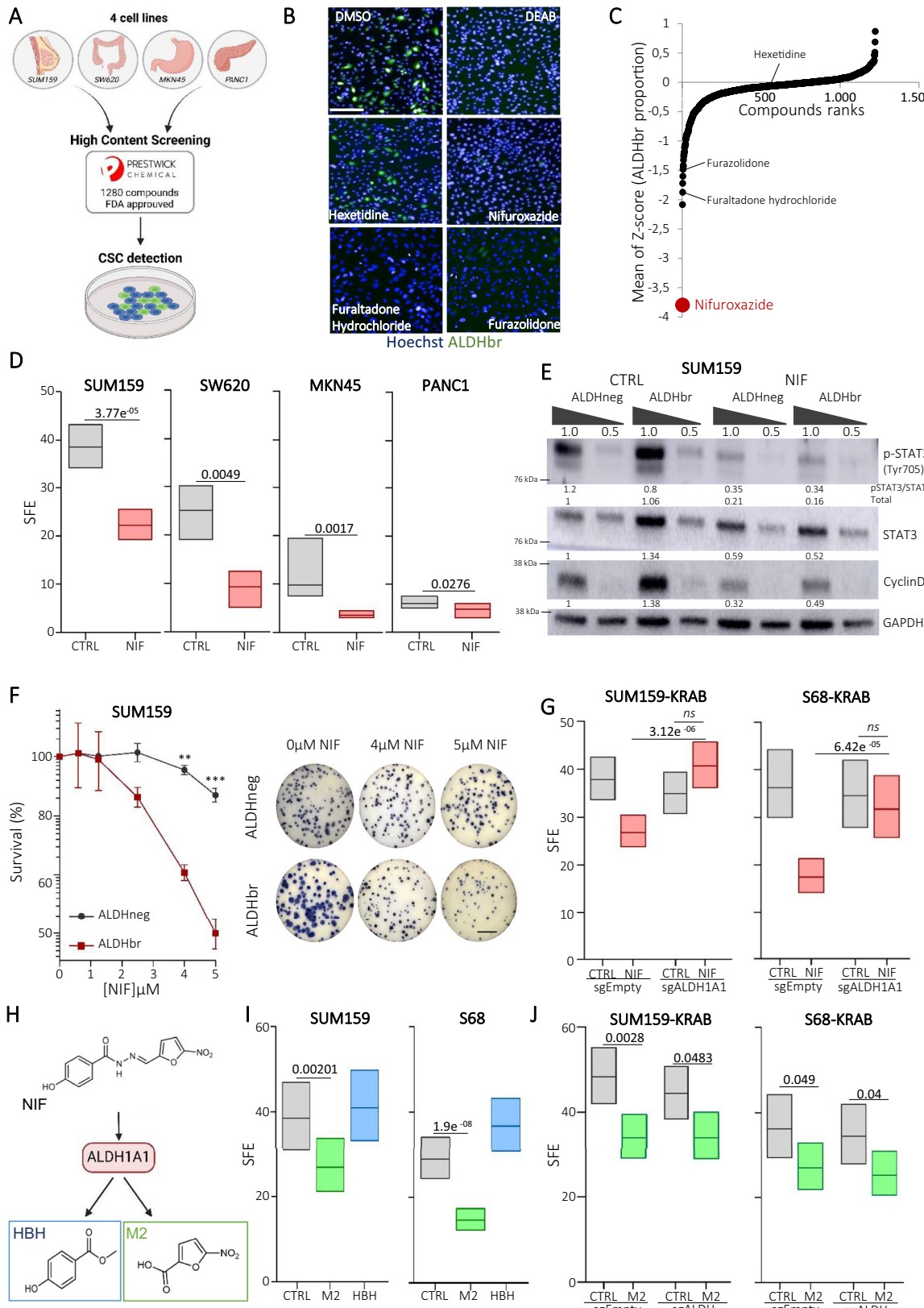

tumorsphere formation (Fig. 1G). These observations suggest that NIF may act as a pro-drug that can be converted into cytotoxic metabolites in breast CSCs that overexpress ALDH1A1. NIF is known to generate two main metabolites: 5-Nitro-2-furaldehyde (M2) and 4-hydroxybenzhydrazide (HBH) (Fig. 1H)[33,34]. Cells treated with each of these metabolites revealed that only M2 treatment recapitulates the

effect of NIF treatment in the ALDHbr cell population and on tumorsphere-forming efficiency (Supplementary Fig. 1R and Fig. 1I) and that M2 effect was not dependent on ALDH1A1 activity (Fig. 1J). Of note, the aminofuran derivative of M2 (named M7) (Supplementary Fig. 1S) had no effect on the breast CSC population, suggesting that the nitro group is essential to mediate the anti-CSC effect of M2

**Fig. 1 | Drug repurposing screen identifies Nifuroxazide (NIF) as a pro-drug specifically bioactivated in cancer stem cells. A** Schematic representation of high content screening strategy. Created in BioRender.com. https://BioRender.com/b62d917. **B** Representative images of the high-content screening capture from 3 technical replicates. ALDEFLUOR cellular staining (ALDHbr) is represented in green. Nuclei are counterstained in blue (Hoechst). Scale bar: 50 μm. **C** Z-score analysis of data from the screen. Scatter points represent compounds, the y-axis represents the mean of Z-Score of each cancer cell line. Compounds inducing a reduction of the ALDHbr cell proportion have a negative Z-score. **D** Tumorsphere forming efficiency (SFE) of cell lines under treatment with Nifuroxazide (NIF) compared to untreated condition (CTRL). $n = 6$ independent experiments. **E** Western blot of STAT3 and its downstream protein, cyclin D1 in SUM159 ALDHneg- and ALDHbr-sorted population for NIF-treated and untreated conditions (CTRL). The mean intensities are indicated below each band for each condition from 3 independent experiments. **F** SUM159 cells exposed to various concentrations of NIF were subjected to clonogenic survival assays and representative images. Scale bar: 2.5 cm. $n = 3$ independent experiments. **G** SFE of sgEmpty and sgALDH1A1 SUM159-KRAB, S86-KRAB in NIF-treated cells, estimated in a limiting dilution assay. $n = 3$ independent experiments. **H** NIF bio-activation with its derived metabolites M2 and HBH. **I** SFE of SUM159 and S68 under treatment with M2, HBH compared to untreated condition (CTRL). $n = 3$ independent experiments. **J** Tumorsphere forming efficiency (SFE) of sgEmpty and sgALDH1A1 SUM159-KRAB, S86-KRAB under M2 treatment and compared to untreated condition (CTRL). $n = 3$ independent experiments. In (**D, G, I, J**) box represents mean ± margin of error (95% Confidence Interval). Statistical significance was calculated using one-sided chi-squared test or $t$-test. In (**F**), data are shown as mean ± SEM (Standard Error of Mean), according to two-way ANOVA followed by Sidak multiple range test. ns (not significant), $*p < 0.05$, $**p < 0.01$, $***p < 0.001$. Source data are provided as a Source Data file.

(Supplementary Fig. 1T, U). Altogether, our results sustain that NIF is a pro-drug, specifically bio-activated in ALDHbr breast CSC in which it releases a cytotoxic metabolite (M2) responsible for the elimination of this cell subpopulation. However, the mode of action of M2 remains unknown, and we assume that it is not restricted to the inhibition of the STAT3 pathway.

### NIF and its metabolite bind to DNA

To investigate the mechanism of action of NIF and its metabolites in the CSC population, we developed a click chemistry strategy. This chemical reaction is employed in biorthogonal labeling approaches to create chemical probes in situ[35]. We have synthesized clickable derivatives for NIF and its metabolites (M2 and HBH), hereafter named NIF-C, M2-C, and HBH-C, by adding an alkyne group to the different compounds (Fig. 2A). These clickable molecules phenocopied the activities of their parental compound by affecting the ALDHbr cell proportion and the tumorsphere-forming efficiency (Supplementary Fig. 2A, B). This indicates that modification for click chemistry does not change the anti-CSC activity of the compounds. Fluorescent labeling of NIF-C in sorted cells (ALDHneg and ALDHbr) revealed a predominant nuclear localization of this compound following 6 h of treatment (Fig. 2B and Supplementary Fig. 2C). Interestingly, after 72 h of treatment, the proportion of ALDHneg cells presenting a fluorescent-labeling in their nucleus was drastically reduced whereas the nucleus of ALDHbr cells remained labeled. These observations suggest that the metabolites released by ALDH1A1-mediated NIF bioactivation may be more stable in the nucleus than unmetabolized NIF. To test this hypothesis, we evaluated the M2-C localization in both cell subpopulations. We observed a nuclear localization of M2-C that was maintained in both cell subpopulations (ALDHneg and ALDHbr) after 72 h of treatment (Fig. 2C and Supplementary Fig. 2C). This data confirms the long-lasting stability of M2 compared to unmetabolized NIF. Of note, the HBH-C can be detected only after a short treatment time (30 min) before a complete vanishing after 6 h of treatment, providing a rationale for the lack of biological activity of this metabolite (Fig. 2D and Supplementary Fig. 2C, D). Because the nuclear stabilization of M2 may suggest an interaction with genomic DNA, we performed quantitative image-based cytometry (QIBC)[36]. Using this approach, combined with click reaction, we were able to evaluate the binding of our clickable derivatives to DNA. In agreement with our previous observations, we detected, after 6 h of treatment, an increase of the median of fluorescence intensity in the nuclei of cells treated with NIF-C or M2-C compared to the cells treated with HBH-C or the untreated conditions (CTRL) (Fig. 2E). Moreover, by following the evolution of the median fluorescence intensity in the nuclei of ALDH-sorted cells, we confirmed that NIF-C (or its metabolites) was only detected in the nuclei from ALDHbr cells after 72 h of treatment (Fig. 2F). Collectively, our data suggest that the NIF-metabolite (M2) is accumulated in the

nuclei of cells where it binds to genomic DNA. In the absence of bio-activation by ALDH1A1, NIF is rapidly eliminated from the nucleus, suggesting that it is only its M2 metabolite that can generate bonds with DNA strands.

### NIF generates DNA interstrand crosslink lesions (ICLs) that accumulate in CSCs

The binding of molecules to DNA can impact genome stability by blocking replication or transcription[37]. To investigate the impact of NIF binding to DNA on genome integrity, we conducted experiments involving cells treated with NIF or M2 and assessed the formation of phosphorylated (serine 139) histone variant H2AX (γH2AX), a marker of replication stress and DNA damage[38]. Under control conditions (DMSO), ALDHneg cells displayed a higher count of γH2AX-positive cells compared to ALDHbr cells, aligning with our prior study indicating that ALDHbr cells exhibit less replicative stress than ALDHneg cells[39]. Upon treatment with NIF or M2, we noted a two-fold increase in γH2AX-positive cells, specifically in ALDHbr cells, compared to the control. As anticipated, HBH treatment did not significantly alter the proportion of γH2AX-positive cells in either cell sub-population (Fig. 3A and Supplementary Fig. 3A). Importantly, the rise in γH2AX-positive ALDHbr cells was not linked to the anti-STAT3 activity of NIF, as evidenced by the absence of γH2AX-positive cell accumulation in each subpopulation treated with STAT3 pathway inhibitors (Supplementary Fig. 3B).

Concurrent with the increase in γH2AX foci, ALDHbr cells treated with NIF, but not ALDHneg cells, exhibited G2/M arrest and a notable increase in polyploid cells (Fig. 3B–D and Supplementary Fig. 3C, D). These observations correlated with heightened expression of G2/M cell cycle-related genes in NIF-treated ALDHbr cells compared to untreated ALDHbr cells (Supplementary Fig. 3E). Collectively, these findings suggest that the NIF metabolite generated in ALDHbr cells can bind to DNA, causing DNA damage, ensuing G2/M arrest, and the production of polyploid cells. Consequently, a significant rise in apoptotic cells was specifically observed in NIF-treated ALDHbr cells, elucidating the reduction in CSC proportion under NIF treatment (Fig. 3E). This cellular response to NIF treatment correspond to the hallmarks of DNA cross-linking induced by genotoxic agents generating DNA mono-adducts and/or interstrand crosslinks (ICLs)[40,41].

To delve deeper into the nature of DNA lesions generated by the NIF metabolite, we utilized two clones of HeLa cells lacking either *ERCC1* (ERCC1[KO])[42] or *FANCD2* gene (FANCD2[KO])[43], two proteins crucial for DNA damage repair and engaged by the Fanconi anemia (FA) pathway[44] (Fig. 3F). FANCD2 is predominantly recruits other proteins involved in ICL repair, such as the homologous recombination machinery (HR)[45,46]. While FANCD2 is necessary only for the repair of ICL but not mono-adduct, ERCC1 is essential for both DNA ICLs repair and mono-adduct DNA excision[47]. M2 treatment led to a dose-

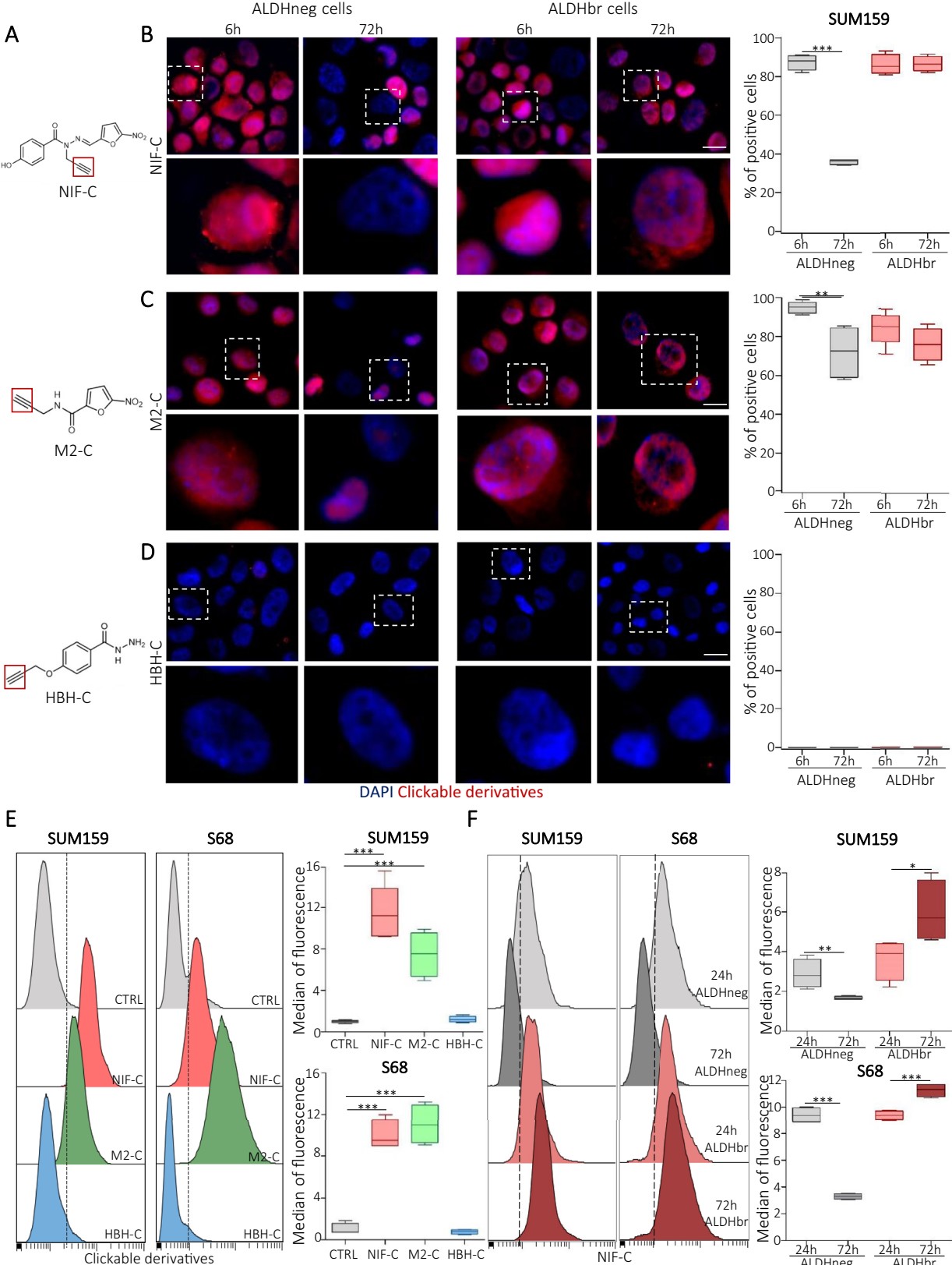

dependent reduction in colony numbers of both FANCD2[KO] and ERCC1[KO] HeLa clones compared to HeLa WT cells (Fig. 3G). These observations suggest that bio-activated NIF primarily generates DNA ICLs, which cannot be repaired in both cells deficient for *ERCC1* or *FANCD2*.

To validate NIF capacity to induce ICLs, we employed a modified alkaline comet assay previously used for detecting DNA ICLs in cells[48]. Essentially, ionizing radiation (IR) treatment induces double-strand breaks (DSB), leading to DNA fragment accumulation and an increase in comet tail moment. Pre-treatment with an ICL-inducing agent causes

**Fig. 2 | NIF and its bioactivated metabolite (M2) binds to DNA. A** Chemical structure of click compounds. **B–D** Detection of NIF-Click (NIF-C), M2-Click (M2-C), and HBH-Click (HBH-C) (red staining) in ALDHneg and ALDHbr SUM159 cells after 6 and 72 h of treatment. Nuclei are counterstained with DAPI (blue staining). For each panel, bottom line represents an enlargement of the area delimited by dashed line. Scale bar: 5 μm. Box plots represent the proportion of cells with a positive click-labeling after different times of treatment. $n = 3$ independent experiments. **E** Detection of indicated click compounds by quantitative image-based cytometry (QIBC) after 6 h of treatment in SUM159 and S68 cells. Data are represented in

boxplots for each click compound, $n = 6$ independent experiments. **F** Detection of NIF-C by QIBC after 24 or 72 h of treatment in ALDHneg or ALDHbr SUM159 and S68 cells. Data are represented in boxplots for each condition $n = 3$ independent experiments. In (**B–D**), boxplots represent median and quartile and whiskers minimum to maximum. In (**E–F**), boxplots represent the median of fluorescence and quartiles and whiskers minimum to maximum. ns (not significant), *$P < 0.05$, **$P < 0.01$, and ***$P < 0.001$ according to one-way ANOVA followed by Dunn's multiple comparison test (**B–D**) or two-way ANOVA followed by Sidak multiple range test (**E–F**). Source data are provided as a Source Data file.

a decrease in the IR-induced tail moment owing to the formation of bridges between DNA fragments (Fig. 3H)[43]. As a positive control, we treated FACS-sorted cells with the ICL-inducing agent melphalan[43,49], which significantly reduced tail moment in both ALDHbr and ALDHneg cells compared to the untreated condition. Intriguingly, NIF-treated cells displayed a substantial decrease in the IR-induced tail moment, specifically in the ALDHbr population, while no effect was observed in ALDHneg cells (Fig. 3I, J and Supplementary Fig. 3F, G).

In summary, our findings demonstrate that NIF acts as an ICL-inducing agent, generating these detrimental DNA lesions specifically upon metabolism in ALDHbr CSCs.

## NIF inhibits the STAT3/Fanconi anemia axis

Accumulating evidence suggests that breast CSCs display constitutive activation of DNA repair systems, making this cell population resistant to genotoxic agents[39]. In this context, our results appear somewhat contradictory with NIF specifically inducing DNA ICL lesions in CSC with a cytotoxic effect. We, therefore, hypothesized that CSCs may exhibit a defect in DNA ICL repair when treated by NIF. Because the monoubiquitination of FANCD2:FANCI heterodimer is a critical step in the activation of the FA pathway in response to ICL-induced replication stress, we first assessed the modification of FANCD2 and FANCI proteins by western blot. As expected, melphalan treatment induced a significant increase in FANCD2 and FANCI monoubiquitination associated with an increase in γH2AX. Surprisingly, cells treated with M2 alone or in combination with melphalan did not present any increase in the monoubiquitinated form of FANCD2:FANCI but rather a significant decrease of these proteins expression (including both the mono-ubiquitinated and non-monoubiquitinated forms); while we did observe an increase in γH2AX protein testifying to the presence of DNA lesions (Fig. 4A). This data may suggest that M2-treated cells are unable to activate the FA pathway to repair ICLs. To test this hypothesis, we performed an ICLick assay[43]. This approach allows the quantification of DNA ICL lesions using click-melphalan as a functional probe for in situ labeling (Fig. 4B). We first silenced *FANCD2* in SUM159 cells using shRNA constructs and a non-targeting shRNA as control (shCTRL) (Supplementary Fig. 4A). Then, using QIBC combined with ICLick we did observe a significant reduction in the median of fluorescence intensity in SUM159 shCTRL cells after 72 h of treatment compared to cells treated 24 h, indicating that DNA ICLs induced by click-melphalan are progressively repaired (Fig. 4C). On the other hand, in SUM159 shFANCD2 cells, click-melphalan lesions were still detected after 72 h of treatment confirming that FANCD2 is required to repair DNA ICLs. Similar results were obtained for SUM159 shCTRL cells treated with M2, demonstrating that M2 inhibits DNA ICL repair. Altogether, these observations suggest that NIF, when metabolized in ALDHbr breast CSC, may be a double-acting drug that first induces DNA ICLs and then inhibits the FA pathway responsible for ICL repair. To identify the protein target that may link NIF effect to FA pathway inhibition, we took advantage of the recently published proteome-wide atlas of drug mechanism of action[50]. This atlas used protein-protein and compound-compound correlation networks to uncover the mechanisms of action for 875 compounds. We interrogated this atlas to identify targeted therapies capable to mimic the effect of NIF

on *FANCI* expression. The top compound list was significantly enriched in MDM2/P53 and JAK/STAT pathway inhibitors (Fig. 4D). While the interaction between P53 and FA pathway has already been established[51], this analysis suggests a role of the JAK/STAT pathway in regulating *FANCI* expression. Given the anti-STAT3 activity of NIF (Fig. 1E), we hypothesized that STAT3 transcription factor activity might regulate *FANCI* gene expression. By performing a motif enrichment analysis, we identified an overrepresentation of STAT3 binding motifs in the promoter of FA genes (Supplementary Fig. 4B, C). We next perform a STAT3 CUT&RUN-qPCR using two sets of primers across the STAT3 binding motif of FANCI promoter and as a positive control two set of primers across the STAT3 promoter itself (Supplementary Fig. 4D). As expected, we measured an enrichment of STAT3 on its own promoter following 1 h of IL-6 (an activator of the JAK/STAT pathway) stimulation (Fig. 4E). This binding of STAT3 to its own promoter is rapid and transient with a return to the baseline (observed in untreated cells and defined by the cells with *STAT3* silencing) after 6 h of IL-6 stimulation. Remarkably, similar results were obtained for the *FANCI* promoter, which exhibited transient STAT3 binding after IL-6 stimulation. To assess whether STAT3 binding to *FANCI* promoter has an impact on *FANCI* transcriptional activity we used a reporter gene assay with the luciferase gene under the control of the *FANCI* promoter (Fig. 4F). As a positive control, we treated SUM159 cells with melphalan which induced a significant increase of the relative luciferase activity reflecting a strong transcriptional activity of *FANCI* promoter. At the opposite, M2 treatment reduced *FANCI* promoter activity, further demonstrating that this compound impacts the transcriptional regulation of *FANCI*. The IL-6 stimulation was enough to induce a relative luciferase activity equivalent to melphalan treatment, whereas the use of STAT3i significantly decreased *FANCI* promoter activity. These results further support that STAT3 regulates the *FANCI* gene expression. Because several FA pathway genes present a STAT3 motif, we evaluated their gene expressions following IL-6 stimulation. All the FA pathway genes tested presented a significant increase of their gene expression, similar to the one observed for *STAT3* (Fig. 4G). Moreover, M2 treatment was able to prevent this IL6-induced overexpression of FA pathway genes. Collectively, these data demonstrated that STAT3 serves as a transcription factor for FA pathway genes and that NIF treatment may abrogate FA pathway repair activity through STAT3 inhibition. To provide functional evidence that STAT3 inhibition is sufficient to render cells deficient in FA pathway activity, we tested the melphalan sensitivity of breast cancer cells invalidated for STAT3 expression versus cells invalidated for FANCD2 expression (Supplementary Fig. 4E). As expected FANCD2-deficient cells are profoundly sensitive to melphalan and STAT3-deficient cells showed similar sensitivity (Fig. 4H and Supplementary 4F). We also carried out an epistasis-type experiment by depleting FANCD2 from STAT3-deficient cells. Cells deficient in both STAT3 and FANCD2 were as sensitive to melphalan as STAT3-deficient cells, confirming that STAT3 is part of FA pathway signaling. The DNA binding motif of STAT3 being quite similar to that of STAT1[52], we tested the sensitivity of STAT1-deficient cells to melphalan to assess the specificity of STAT3 binding. STAT1 inhibition did not recapitulate the melphalan sensitivity observed in STAT3-deficient cells, confirming the specificity of STAT3 on the regulation of

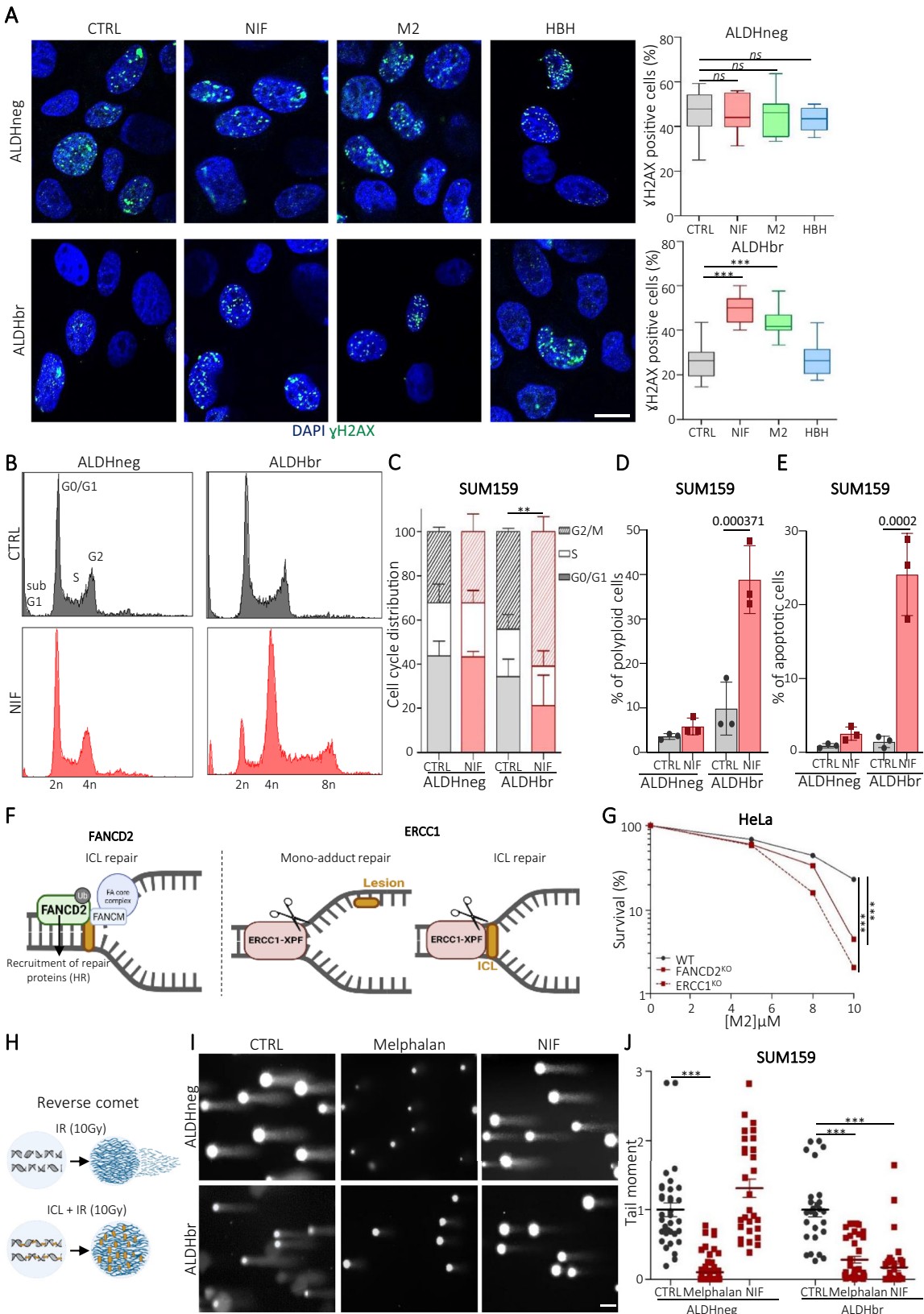

FA pathway (Supplementary Fig. 4G, H). Lastly, we used the ICLick assay to evaluate the DNA ICLs repair capacity of SUM159-KRAB cells depleted for STAT3. We did observe a maintenance of the click-melphalan lesions in the cells STAT3 deficient, attesting the inability of these cells to repair DNA ICLs (Fig. 4I).

In summary, we reported that STAT3 serves as a transcription factor of the FA pathway genes. Inhibition of STAT3 by NIF treatment renders cells deficient in the FA pathway, which, we hypothesize, is at the basis of the cytotoxic effect of NIF in CSC that will accumulate unrepaired DNA ICLs.

**Fig. 3 | NIF induces DNA inter-strand crosslinks lesions that accumulate in breast cancer stem cells. A** Representative images of γH2AX foci (green staining) in ALDHneg and ALDHbr SUM159 cells. Nuclei are counterstained with DAPI (blue staining). Scale bar: 15 μm. Box plots represent the proportion of γH2AX -positive cells for each cell subpopulation under the indicated treatment and compared to the untreated condition (CTRL) for each drug. *n* = 3 independent experiments. Monitoring of cell cycle profile by QIBC in ALDHneg and ALDHbr SUM159 cells exposed to NIF and compared to untreated condition (CTRL) (**B**) Quantification of cell cycle phase distribution in each condition (**C**) Proportion of polyploid cells in each condition (**D**). *n* = 3 independent experiments. **E** Proportion of apoptotic cells determined by annexin V labeling in ALDHneg and ALDHbr SUM159 cells exposed to NIF and compared to untreated condition (CTRL). *n* = 3 independent experiments. **F** Role of FANCD2 and ERCC1 in DNA lesions (ICLs and/or mono-adduct) repair. Created in BioRender.com. https://BioRender.com/b62d917. **G** HeLa WT,

FANCD2$^{KO}$, or ERCC1$^{KO}$ cells exposed to various concentrations of M2 were subjected to clonogenic survival assays. *n* = 3 independent experiments. **H–J** Schematic representation of reverse comet assay principle (**H**), created in BioRender.com. https://BioRender.com/b62d917. Representative images of reverse comet assays conducted in ALDHneg and ALDHbr SUM159 cells subjected to the indicated treatments. Scale bar: 100 μm (**I**). Reverse comet assay quantification, each dot corresponding to tail moment of one comet with the indicated drugs. *n* = 200 comets analyzed. In (**A**), boxplots represent the median of fluorescence and quartiles and whiskers minimum to maximum. In (**B**–**E**, **H**–**J**), data are shown as mean ± SD (Standard Deviation). ns (not significant), *$P < 0.05$, **$P < 0.01$ and ***$P < 0.001$ according to one-sided Fisher test (**A**, **C**–**E**), two-way ANOVA followed by Sidak multiple range test (**G**), one-way ANOVA non-parametric test (**J**). Source data are provided as a Source Data file.

## NIF induces a chemical HRDness that is synthetic lethal with PARPi

FANCA deficiency has been shown to sensitize ovarian and prostate cancers to PARPis, demonstrating the existence of synthetic lethality of FA proteins other than BRCA1/2 with PARPi treatment[53]. Moreover, ATM depletion induces proteasomal degradation of FANCD2 and sensitizes neuroblastoma cells to PARPis[54]. In the case of breast cancer, and based on the clear clinical benefits of PARP inhibition, a large effort is made to propose new clinical strategies to extend PARPi treatment beyond BRCA-mutant carriers toward a broader population of patients presenting tumors with homologous recombination repair deficiency (HRD). While there is an urgent need to identify surrogate markers capable of reliably predicting HRD, a new strategy based on the development of compounds inducing a chemical HRDness could offer the possibility of sensitizing tumors to PARPi regardless of their HRD status[55].

Having shown that NIF treatment induces a deficiency of FA pathway and knowing that BRCA genes are FA pathway players[44], we hypothesized that NIF could be a chemical HRDness inducer targeting breast CSC. We first assessed the variation in HR activity by measuring the formation of RAD51 foci following induction of double-strand breaks (IR treatment) or ICLs (melphalan treatment) in the presence or absence of M2. Both IR or melphalan treatments induced RAD51 foci, and the presence of M2 abolished RAD51 recruitment in melphalan-treated cells, mimicking an HR deficiency (Fig. 5A). These observations are in line with a previous study that reported an FA-independent HR induction following IR[56], whereas FA is required to promote HR following ICLs. These data sustain that NIF treatment leads to HRD induction in cancer cells and thus suggest that NIF could be synthetically lethal with PARPi. To test this hypothesis, we selected breast cancer cell line with known HR status[57], and confirmed the correlation between HRD and the response to PARPi using clonogenic assays (Supplementary Fig. 5A). In HRP (Homologous Recombination Proficient) cell lines, the ALDHneg cells appeared to be more sensitive to PARPi than the ALDHbr cells whereas no difference was observed in HRD cell line (Fig. 5B, D and Supplementary Fig. 5B). This observation is in agreement with previous studies reporting a relative resistance of breast CSC to PARPi due to an enhance HR activity[39,58]. To evaluate a potential synthetic lethality interaction between NIF and PARPi, we performed a clonogenic assay on sorted cells treated with a non-toxic dose of PARPi and exposed to increasing doses of NIF. In both HRP cell lines, we observed an increased sensitivity of PARPi-treated ALDHbr cells to NIF, whereas this drug combination had no additional effect on the ALDHneg cell survival (Fig. 5C and Supplementary Fig. 5C). In HRD cell line, we did not observe any enhanced effect of the PARPi/NIF combination on the survival of ALDHbr or ALDHneg cells (Fig. 5E). Based on these results, NIF appears to be synthetic lethal with PARPi specifically in breast CSC. To functionally prove the effect of this drug combination on the breast CSC population, we performed a

tumorsphere assay using cells treated with PARPi, NIF alone, or in combination. In HRP cell lines, we did observe a drastic decrease of the tumorsphere-formation efficiency in cells treated with the PARPi/NIF combination compared to cells treated with one compound only (Fig. 5F and Supplementary Fig. 5D). At the opposite, in HRD cell line, PARPi or NIF treatment alone was sufficient to induce a strong effect on the tumorsphere-formation efficiency, with no additional effect of the drug combination (Fig. 5G). Of note, clonogenic assays using HRP cell lines (SUM159-KRAB and S68-KRAB) depleted for STAT3 revealed an increased sensitivity to PARPi alone compared to control cells, further suggesting that the synthetic lethality interaction between NIF and PARPi is mediated in part through the anti-STAT3 activity of NIF (Supplementary Fig. 5E). Using tumorsphere assay, we further confirmed that HRP cells (SUM159-KRAB and S68-KRAB) silenced for STAT3 and treated with PARPi presented a decrease of the tumorsphere-formation efficiency (Supplementary Fig. 5F). However, STAT3 inhibition presented a moderate synergistic effect with PARPi compared to cells treated with the PARPi/NIF combination, suggesting that the generation of DNA ICLs by NIF potentializes its effect on CSC beyond the simple STAT3 blockade. To summarize, NIF treatment, following its bio-activation by ALDH1A1 in breast CSC, produces the metabolite M2 that generates DNA ICLs. In the same time, NIF induces a chemical HRDness via STAT3 inactivation that is sufficient to be synthetic lethal with PARPi, similar to cells with HRD due to BRCA mutations (Fig. 5H). This triple action (DNA ICLs, chemical HRDness, synthetic lethality with PARPi) has a strong impact on the survival of the breast CSC population.

## Synthetic lethality by NIF/PARPi combination in patient-derived xenografts

To validate NIF treatment as an appropriate strategy to sensitize breast CSCs to PARPi, we performed a preclinical assay using triple-negative patient-derived xenografts (PDXs)[59,60]. To mimic clinical conditions, we determine HRDness of PDXs using the SOPHIA DDM® platform employed by oncologists in routine diagnostic testing. We selected PDXs with different predicted HRDness (Fig. 6A). CRCM434 presented a genomic integrity index (GI index) of 15 with deleterious mutations in *FANCA* and *RAD54L*, two key actors of the FA-BRCA pathway, qualifying this PDX as HRD. CRCM494 and Pandora21 did not present any mutation in HR-related genes but presented a positive GI index, reflecting extended genomic scarring and suggesting an acquired HRDness. Pandora7 harbored a negative GI index, predicting this PDX as HR-proficient. We then treated mice bearing these PDXs using Olaparib (PARPi) and NIF alone or in combination versus vehicle-treated mice (Supplementary Fig. 6A). As expected, CRCM434 (*RAD54L*$^{mut}$/*FANCA*$^{mut}$) presented a strong tumor response to PARPi treatment with a complete block in tumor growth (Fig. 6B). NIF treatment alone did not have any effect on tumor growth and its association with PARPi did not potentialize its effect. Surprisingly,

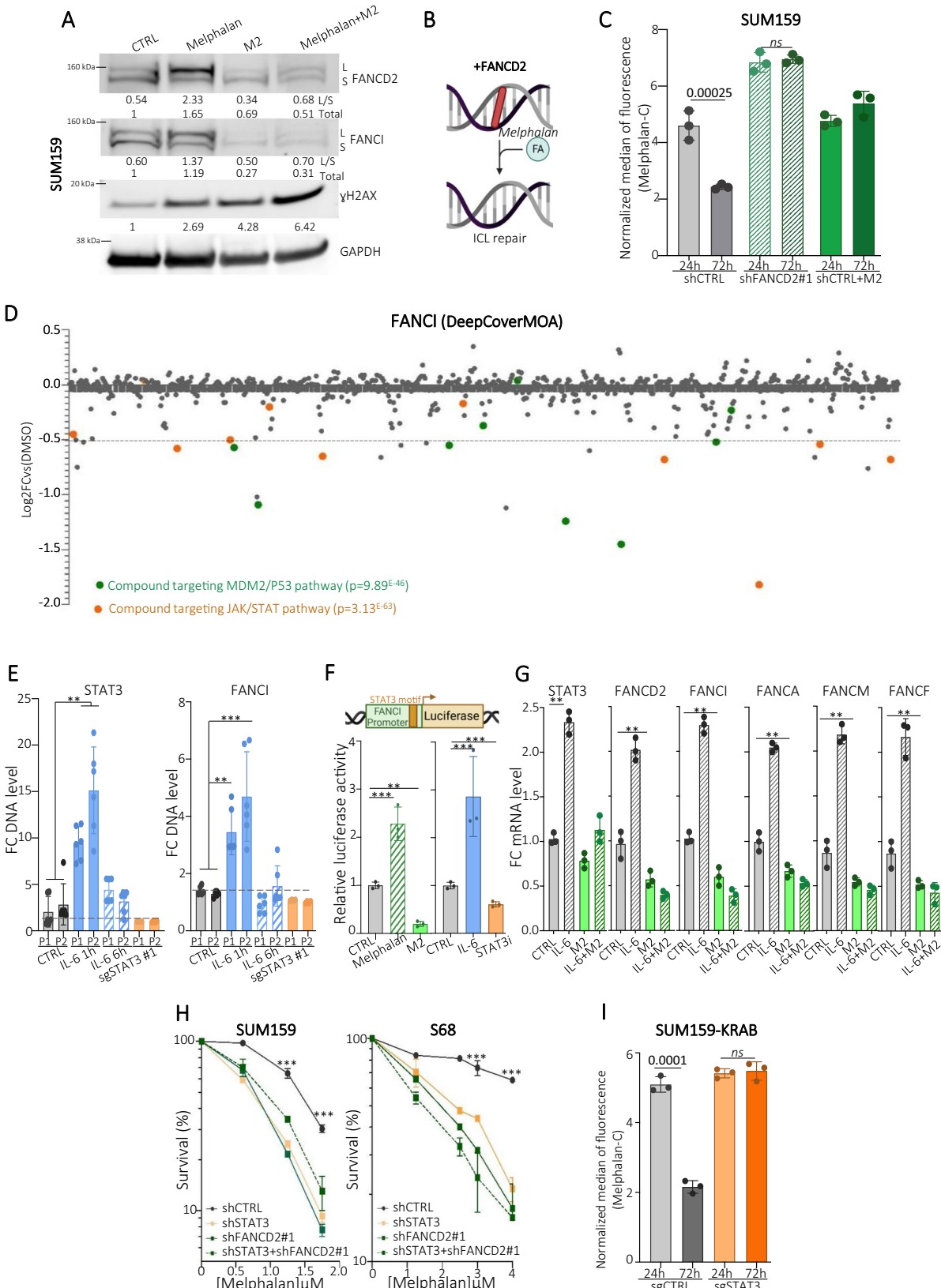

CRCM494 and Pandora21, predicted as HRD, did not present any sensitivity to PARPi treatment, whereas its combination with NIF slowed tumor growth (Fig. 6C, D). As anticipated, for Pandora7 (HRP) PARPi treatment has no effect on tumor growth, however combination with NIF treatment appears to synergize slowing down tumor growth (Fig. 6E). Of note, if the effect of PARPi/NIF combination on tumor growth appears limited, at least in the timeframe of our experiments, it is concordant with previous reports on anti-CSC therapy that selectively targets CSCs while mainly sparing actively dividing differentiated cancer cells, ending up with limited short-term effect on tumor growth[13,23,61]. To detect a potential impact on the tumorigenic cell populations, we performed a limiting dilution assay into secondary

**Fig. 4 | NIF Inhibit the STAT3/Fanconi anemia axis. A** Detection of FANCD2, FANCI, and γH2AX by Western blot in SUM159 cell protein extracts after treatment with melphalan, M2, or the combination. Numbers under the blots indicate the fold induction relative to untreated samples. L/S indicates the ratio of mono-ubiquitinated (L, upper band) to non-monoubiquitinated (S, lower band) FANCD2 or FANCI protein. **B** Schematic representation of ICLick assay. DNA lesions generated by click-melphalan are detected in red, and its repair can be monitored via the disappearance of the click-labeling. Created in BioRender.com. https://BioRender. com/b62d917. **C** Quantification of click-labeling (fluorescence intensity) in shCTRL, shCTRL+M2, and shFANCD2 SUM159 cells treated with click-melphalan after 24 h and 72 h of treatment. **D** Expression of FANCI following the individual treatment of 875 compounds and normalized with DMSO-treated condition used as control. This dot plot has been generated with the DeepCoverMOA web interface. Two families of compounds that significantly decrease FANCI expression are highlighted with MDM2/P53 inhibitors in green or JAK/STAT pathway inhibitors in orange.

**E** CUT&RUN-qPCR of STAT3 in SUM159-KRAB cell under indicated treatment. Data are represented by the fold-change of DNA level normalized on sgSTAT3 condition. P1 and P2 correspond to the two sets of primer designed on *STAT3* and *FANCI* promoter sequence. *n* = 6 independent experiments. **F** Schematic representation of FANCI reporter (top panel). Relative luciferase activity measured in SUM159 cells expressing the FANCI promoter after indicated treatment. **G** mRNA levels of FA genes, measured by qRT-PCR, in SUM159 treated with M2, IL-6, or in combination, normalized with untreated conditions (CTRL). **H** SUM159 and S68 shCTRL, shFANCD2#1, and shSTAT3 cells exposed to various concentrations of melphalan were subjected to clonogenic survival assays. **I** Quantification of fluorescence intensity in sgCTRL and sgSTAT3 SUM159-KRAB treated with click-melphalan and measured after 24 h and 72 h of treatment. In (**C, E–I**), data are shown as mean ± SD. ns (not significant), *P* < 0.05, **P* < 0.01, and ***P* < 0.001 according to one-sided Fisher test (**C, E, F, G, I**), two-way ANOVA followed by Sidak multiple range test (H). *n* = 3 independent experiments. Source data are provided as a Source Data file.

mice. For CRCM434 (HRD), residual cells isolated from PARPi-treated tumors (alone or in combination with NIF) presented a significant reduction in the tumor-initiating capacity in secondary mice compared with control and NIF alone (Supplementary Fig. 6B). Limiting dilution transplantation analysis further confirmed that the abundance of tumorigenic CSCs is significantly lower in PARPi-treated tumors than in control or NIF-treated tumors (Fig. 6F). In line with prior evidence in tumors with mutations in HR-related genes, PARPi is sufficient to induce a massive reduction of tumorigenic cells[62]. In both PDXs HRD-predicted but resistant to PARPi (CRCM494 and Pandora21), residual cells isolated from PARPi-treated tumors presented a similar tumor-initiating capacity in secondary mice than in control. However, residual cells from NIF/PARPi-treated tumors presented a significant decrease of the tumor-initiating capacity (Supplementary Fig. 6C, D). This observation was confirmed by the strong reduction of tumorigenic CSCs in residual cells treated by the combination of NIF with PARPi compared with all other treatment arms (Fig. 6G, H). Similar observations were done in Pandora7, predicted HRP and PARPi-resistant (Supplementary Fig. 6E and Fig. 6I). Taken together, our data demonstrate that NIF treatment can sensitize CSCs to PARPi in tumors initially PARPi-resistant and independently of their HRD score. It strongly suggests that NIF acts as a chemical inducer of HRDness. To further test this hypothesis, we assessed the ability of each PDX to repair DNA damage under NIF treatment. We first performed an ICLick assay on patient-derived organoids generated from each PDX (PDXOs). DNA lesions generated by melphalan-C were maintained in CRCM434, whereas it was, at least partially, repaired in all the others PDXOs, providing a functional explanation of PDX treatment response to PARPi (Fig. 6J–M). We then adapted the ICLick assay by substituting melphalan-C by M2-C to test the capacity of PDXOs to repair DNA lesions under M2 treatment. None of the PDXOs were capable of repairing M2-induced DNA lesions, further demonstrating the potential of NIF to induce a chemical HRDness, even in cells initially PARPi-resistant. Altogether, these observations pave the way to new therapeutic opportunities for HR-proficient patients or those with unknown HRD status.

## Discussion

Our study highlights the strong potential of developing drug repurposing strategies to identify new anti-CSC therapies. Here, we identify an antibiotic with known intestinal antiseptic properties as a new compound that efficiently decreases the tumorigenic cell population. In this context, non-oncology drugs represent a "treasure trove" for the identification of original compounds suitable for cancer treatment.

The new uses for approved drugs that are outside the scope of their original medical indication imply identifying new modes of action of repurposed drugs. Our work provides mechanistic insights into how

NIF works at a molecular level. Indeed, in breast cancers, contrary to what has been reported in melanomas[32], ALDH inhibition alone is not sufficient to explain the anti-CSC effect of NIF, but it rather depends on the inhibition of STAT3-FA signaling coupled with the generation of DNA ICLs. We identified STAT3 as non-canonical transcription factor of the FA machinery. STAT3 has been largely described to be a key factor in cell cycle progression by controlling G1/S transition through the regulation of cyclin D1[63]. It may also influence the S/G2 transition by controlling the activation of ATR-CHK1 and ATM-CHK2 signaling[64]. Here, we propose that STAT3 may be a central node orchestrating cell cycle progression and the coordinated activation of DDR signaling. Therefore, STAT3 may serve as mechanism to secure DNA integrity prior to complete cell cycle. It may be particularly activated during inflammation to protect tissue from pro-inflammatory cytokines with DNA-damaging activities. As an example, in pancreas with local inflammation, β-cells are exposed to IL1β and IFNγ secretion that causes nitric oxide-mediated DNA damage. In this inflammatory environment, the secretion of IL-6 contributes to the activation of STAT3 signaling and prevents the accumulation of DNA damages[65]. Based on our result, we can suspect that STAT3 activates the FA-BRCA pathway that enables β-cell survival and growth in an inflammatory environment. In a tumoral context, cancer cells are exposed to an exacerbated replicative stress-inducing genome and chromosomic instability. To survive, cancer cells must tolerate and eventually limit the toxic stresses imposed by aneuploidy. Recent report demonstrated that breast cancer cells with chromosomic instability rely on activation of inflammatory signaling mediated by cGAS and STING to survive. It was demonstrated that cGAS triggers IL-6 induction, which in turn activates STAT3-dependent pro-survival signaling[66]. Based on our result, we can suggest that the activation of cGAS–STING–IL-6–STAT3 axis may also trigger FA-BRCA machinery to limit replication stress and promote cell survival. Taken together, these studies exposed a targetable vulnerability for cancer cells with high genomic instability, with STAT3/FA-BRCA pathway being a central component of this signaling cascade. Our findings furthermore suggest that genotoxic agents in combination with STAT3/FA-BRCA pathway inhibitors could be an effective treatment for breast cancers. Due to its dual effect, NIF appears to fulfill both actions by inducing FA pathway deficiency and generating DNA ICLs.

In addition to the aforementioned potential of NIF to be used as a single agent inducing two synergistic toxic effects, we tested its potential to be synthetic lethal with PARPi. Clinical evidence demonstrated the profound sensitivity to PARPi of tumors from patients with BRCA mutations[53,67,68]. The search for combination therapies that would result in impaired HR with subsequent sensitization to PARPis in cells with efficient HR repair (or even unknown HR status) has been a great deal of interest from researchers and clinicians. Besides DDR inhibitors such as those that target RAD51,

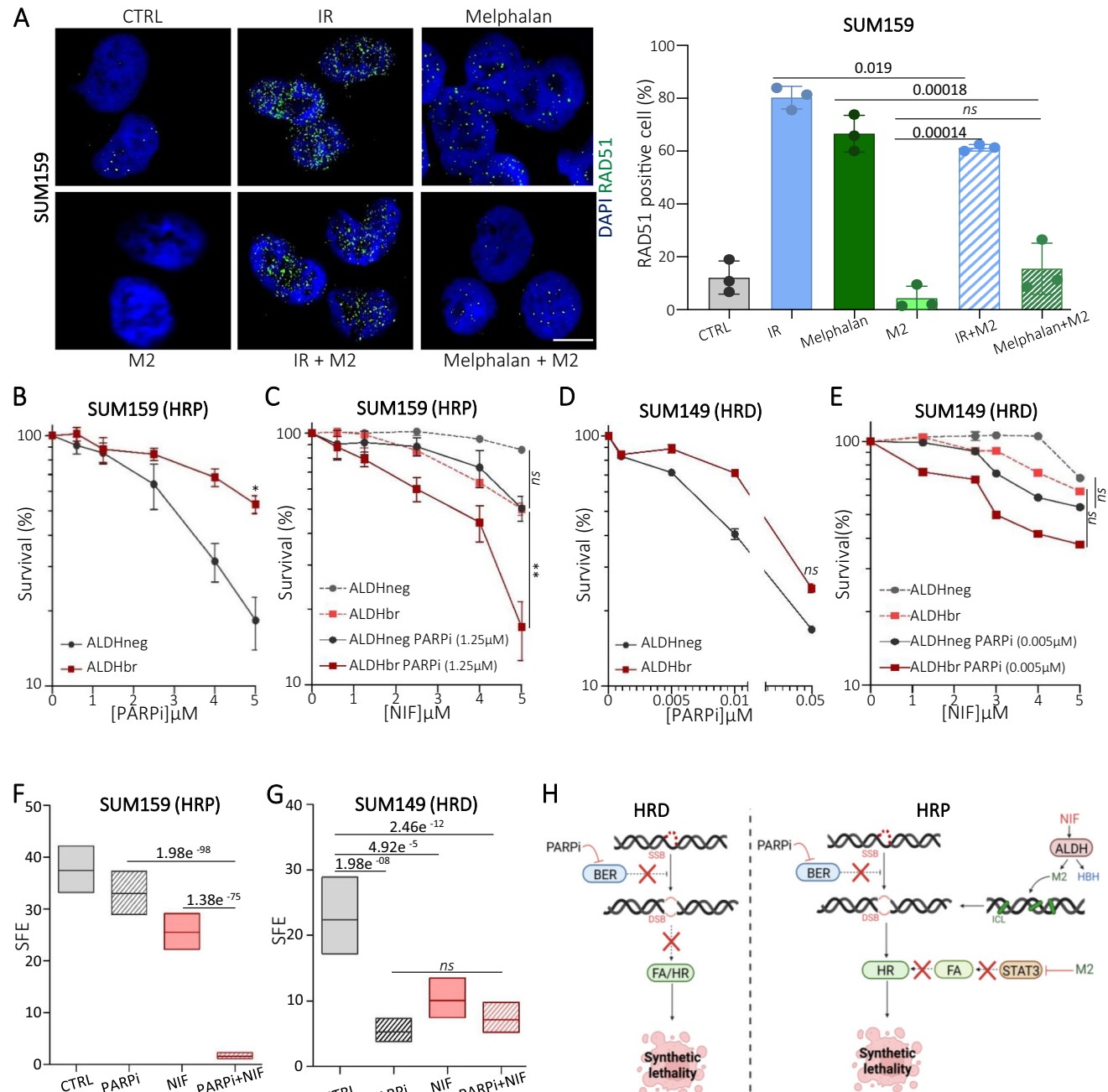

**Fig. 5 | NIF is an inducer of chemical HRDness. A** Representative images of RAD51 foci (green staining) in SUM159 cells irradiated (IR) or treated with melphalan alone or in combination with M2 and compared to untreated condition (CTRL). Nuclei are counterstained with DAPI (blue staining). Scale bar: 5µm. Box plots represent the proportion of RAD51-positive nuclei for each treatment condition. **B–E** SUM159 (HRP) and SUM149 (HRD) ALDHbr and ALDHneg cells exposed to various concentrations of PARPi were subjected to clonogenic survival assays (**B, D**). SUM159 and SUM149 ALDHbr and ALDHneg cells treated with a sublethal dose of PARPi (or untreated cells) were exposed to various concentrations of NIF and subjected to clonogenic survival assays (**C, E**). **F–G** SFE of SUM159 and SUM149 NIF-, PARPi- or combination-treated cells. **H** Working model illustrating chemical HRDness of NIF and its synthetic lethal interaction with PARPi in HRP cells compared to synthetic lethality induced in HRD cells. Created in BioRender.com. https://BioRender.com/b62d917. In (**A–E**), data are shown as mean ± SD, and in (**F–G**) box represents mean ± margin of error (95% Confidence Interval) of 3 independent experiments. ns (not significant), *$P < 0.05$, **$P < 0.01$ and ***$P < 0.001$ according to t-test (A), two-way ANOVA followed by Sidak multiple range test (**B–E**), One-sided Chi-squared test (**F, G**). Source data are provided as a Source Data file.

ATR, CHK1/2, or WEE1, a class of "chemical HRDness" inducers is described[55]. These compounds mainly target oncogenic pathways with a focus on the PI3K/AKT/mTOR signaling. If the underlying mechanism has not been firmly identified, it seems that PI3Ki-treated cells become more dependent to PARP activity due to a down-regulation of BRCA1/2 gene expression[69]. Similar results have been obtained with mTORi that suppress HR repair in BRCA-proficient

TNBCs[70] or with MEKi in tumors with RAS mutant[71]. Because the scientific community is facing difficulty to propose efficient biomarkers predicting HRD beyond the detection of mutations in HR genes, the use of chemical HRDness strategies may be a unique opportunity to maximize the number of individuals who may benefit from PARP inhibition. Although we might assume greater toxicity of therapeutic strategies combining chemical HRDness with PARPi in

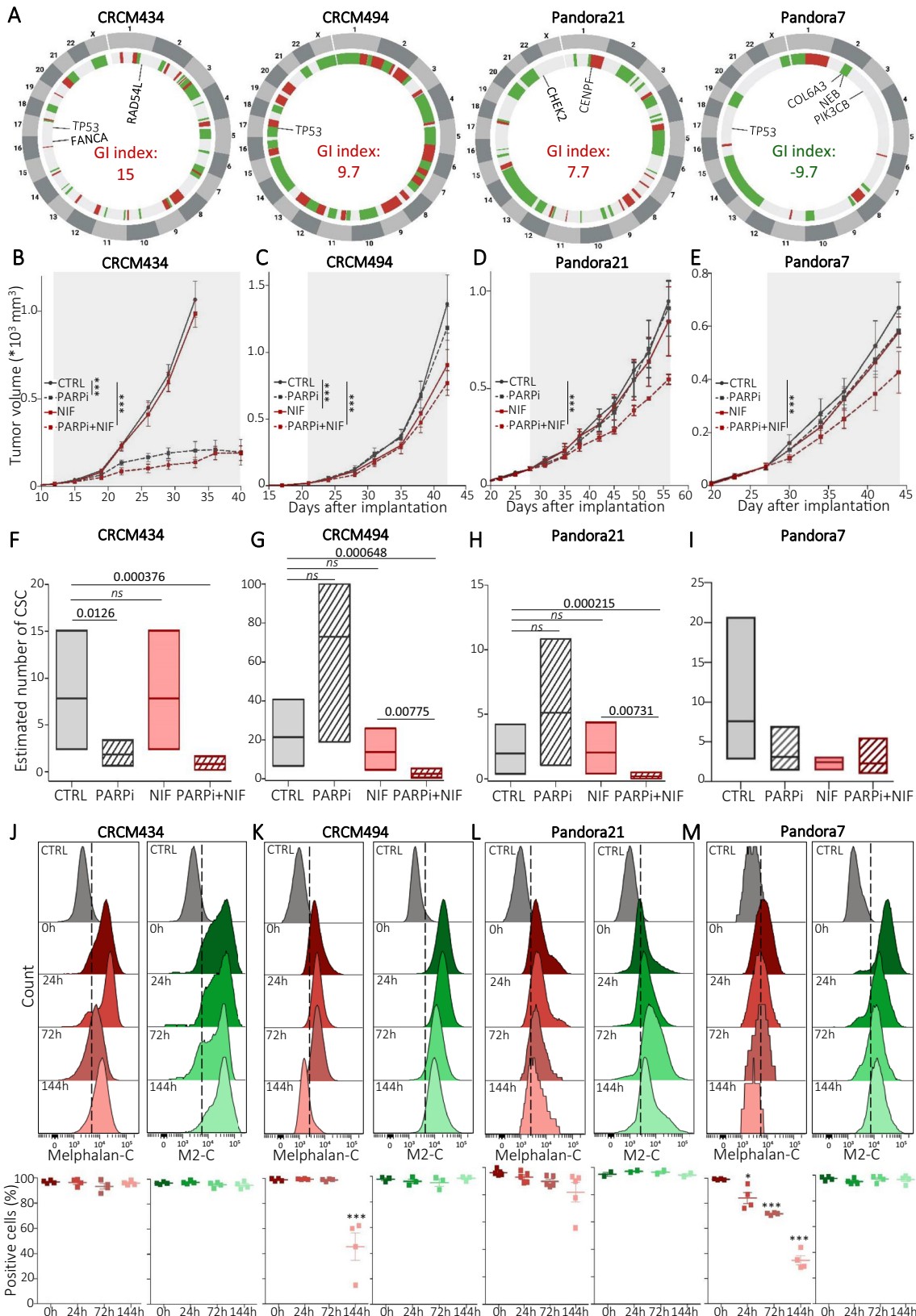

tumor cells with high genomic instability, the adverse-event profile will have to be carefully evaluated. In this context, our study provides rationale for translation of the therapeutic combination of NIF with PARPis into clinical trials targeting BRCA-proficient TNBC patients and potentially patients with tumors of other tissues (such as ovarian cancer).

## Methods

### Ethics statement

Samples of human origin and the associated data were obtained from the IPC/CRCM Tumor Bank, which operates under authorization # AC-2013-1905 granted by the French Ministry of Research. Prior to scientific use of samples and data, patients were appropriately

**Fig. 6 | NIF/PARPi combination induced synthetic lethality in PARPi-resistant PDXs. A** Circos plot showing genomic variations in PDX models. Color-coded chromosomes are arranged around the outside of the circle; CNA is shown in the inner ring (green, deletion; red, gain). Mutations and Genomic Instability index (GI index) are located inside the circus plot. **B**–**E** Effect of NIF and PAPRi treatment alone or in combination on the tumor growth of each PDX, compared with the vehicle-treated condition (CTRL). The gray area corresponds to the treatment period. Growth curves represent the mean ± SEM of tumor volume for $n = 8$ tumors per treatment group. **F**–**I**. Box plots represent bCSC frequency calculated using an extreme limiting dilution analysis (ELDA). The results are expressed as the estimated number of bCSCs for 100 tumor cells. $n = 15$ injections per PDX.

**J**–**M** Detection of click-melphalan (Melphalan-C) or click-M2 (M2-C) residual DNA lesions after different time points of treatment in patient-derived organoids (PDXOs) (upper panel). Dot plots represent the percentage of fluorescent positive cells for Melphalan-C (in red) and M2-C (green) at different time points and for each PDXO. $n = 4$ independent experiments. In (**F**–**I**) box represents mean ± margin of error (95% Confidence Interval), and in (**J**–**M**), data are shown as mean ± SD. ns (not significant), *$P < 0.05$, **$P < 0.01$, and ***$P < 0.001$ according to two-way ANOVA followed by Bonferroni correction (**B**–**E**, **G**), one-sided Chi-squared test (**F**–**I**), and two-way ANOVA followed by sidak multiple range test of 3 independent experiments (**J**–**M**). Source data are provided as a Source Data file.

informed and filed a written consent, in compliance with French and European regulations. The experiments conformed to the principles set out in the WMA Declaration of Helsinki and the Department of Health and Human Services Belmont Report. Animal studies were conducted in agreement with the French Guidelines for animal handling and were approved by local ethics committee (Agreement no. #16487-2018082108541206 v3). Breast cancer has a large female predominance, and all of the PDX models were generated from patients self-identified as female. Due to biological differences, including the influence of hormones, only female mice were used in the PDXs. We used (6–8 weeks) NOD. Cg-Prkdcscid Il2rgtm1Wjl/SzJ mice (Ref: 005557, Charles River). Mice were housed under sterile conditions with sterilized food and water provided *ad libitum* and maintained on a 12-h light and 12-h dark cycle, temperatures of 19–21 °C with 40–60% humidity. Of note, mouse weight loss > 20%, tumor necrosis, tumor volume > 1500 mm³, ruffled coat + hunched back, weakness, and reduced motility were monitored daily and considered as endpoints.

### Cell culture
HT29, SW620, and PANC1 come from ATCC (https://www.atcc.org/, catalog number: HT-38, CCL-227, and CRL-1469 respectively). MKN45 comes from Cytion (#300489). CRC1 cells were established from a CRC biopsy (CHU-Carémeau, Nîmes, France, ClinicalTrial.gov Identifier#NCT01577511), as previously reported[72]. SUM159 and SUM149 were given by Dr. S.Ethier (Karmanos Cancer Center, Detroit, MI, USA), and S68 was given by Dr. V. Castros (Université de Rennes, France). CRC1, SUM159, SUM149, and S68 were gifted by the labs that originally established these cell lines and were, therefore, not subjected to authentication. All cell lines were grown in the standard medium as previously described[27]. HeLa (WT, FANCD2KO, and ERCC1KO) were given by C. Lachaud (CRCM, Marseille)[43] and cell were maintained at 37 °C in DMEM (Dulbecco's modified Eagle's medium; Gibco) supplemented with 10% fetal bovine serum (FBS) and 1% Antibiotic-Antimycotic (Gibco, 15240-062). *Mycoplasma* contamination was excluded by MycoAlert PLUS assay (LONZA, LT07-318). According to the manufacturer's instructions, all the cell lines presenting a ratio < 1 are considered mycoplasma negative.

To silence *STAT3*, *STAT1* and *FANCD2*, SUM159 and S68 cells were transduced with lentiviral constructs: shSTAT3: pSMART-hCMV/TurboRFP (5′-AGCTGGAACAGATGCTCAC) (Horizon #V3SH7590-225360 815) shSTAT1: pSMART-hCMV/TurboRFP (5′-CGAACATGACCCTATC ACA) (Horizon #V3SH7590-226568750) shFANCD2#1: pSMART-hCMV/ TurboGFP (5′ TGGTCCATCAACTACACCG) (Horizon #V3SH7590-22 8098457) shFANCD2#2: pSMART-hCMV/TurboGFP (5′ ACAATGAA-CAATTTCTTGG) (Horizon #V3SH7590-225852447) shCTRL: SMART-vector Lentiviral controls hCMV (Horizon #S-005000-01).

### CRISPRi
To silence *STAT3*, SUM159, S68 and SW620 cells were co-transduced with two lentiviral constructs: pHR-SFFV-KRAB-dCas9-mCherry (Addgene Plasmid #60954,[30] and pU6-sgRNA EF1Alpha-puro-T2A-BFP) (Addgene Plasmid #60955)[30], the latter encoding a control (5′-GCG

CCAAACGTGCCCTGACGG) or *STAT3*-targeting sgSTAT3#1 (5′-GGTT CCGACGTCGCAGCCGA) and sgSTAT3#2 (5′-AACAAGCCCCAACCG-GATCC). Lentiviral infection was conducted by plating 250.000 cells on 6-well plates and incubating them overnight (o/n) with 1 mL of culture medium, polybrene (8 μg/mL), and 5–10 μL of lentivirus (MOI = 2). The cells were then washed twice with PBS and expanded in their usual culture medium. Cell sorting was performed with a FACS Aria III instrument (BD Biosciences) to enrich mCherry and BFP double-positive cells.

To silence *ALDH1A1*, SUM159/S68-KRAB cells were transduced with the lentiviral vector pLKO.1-blast-U6-sgRNA-BfuA1-stuffer encoding 2 sgRNAs per target gene (ALDH1A1 sgRNA1: 5′-TGATTCGGCTC CTGGAACAC and sgRNA2: 5′-AGGTAAGTCTGGCGTGCCTG) as previously described[73].

### Drugs
Cell lines were continuously treated in adherent conditions with Nifuroxazide (NIF) (stock concentration SC = [10 mM], Selleckchem, S4182), 5-Nitro-2-furaldehyde (M2) (SC = [20 mM], TCI, N0387), 4-Hydroxydenzhydrazide (HBH) (SC = [20 mM], Alfa Aesar, A12702), Methyl 5-amino-2-furoate (M7) (SC = [20 mM], Alfa Aesar, L05958), Olaparib (PARPi) (SC = [10 mM], Selleckchem, S1060), Melphalan (SC = [20 mM] Merck, M2011), Napabucasin (STAT3i) (SC = [10 mM] TargetMol, T3218), Ruxolitinib (JAK2i) (SC = [10 mM], Selleckchem, S1378), NIF-C, M2-C and HBH-C (SC = [10 mM]), Melphalan-C (SC = [10 mM], from C. Lachaud laboratory, CRCM, France)[43]. All compounds were resuspended in dimethyl sulfoxide (DMSO, Sigma). The final concentrations used for each drug to treat each cell line model are reported in Supplementary Table 1. IL-6 human (Sigma, SRP3096) resuspended according to the manufacturer's instruction. For the in vivo experiments, NIF [300 mg/kg] (Euromedex, TA-T1563) and Olaparib [50 mg/kg] were resuspended in a solution of 25% cremophore/ethanol.

### Clickable nifuroxazide analogs synthesis
All solvents and chemicals were purchased from commercially available sources and used without further purification or purified according to Purification of Laboratory Chemicals (Armarego, W.L.F.; Chai, C.L.L. 5th Ed.). Solvents were dried under standard conditions. Reactions were monitored by thin layer chromatography (TLC) using pre-coated silica on aluminum plates from Merck (60F$_{254}$). TLC plates were visualized with UV-light and/or by treatment with ceric ammonium molybdate solution (CAM) or ninhydrin solution and heating. Products were purified on column chromatography with Silica gel 60 from Macherey Nagel (0.036-0.071 mm; 215–400 mesh), a CombiFlash Rf+ Teledyne Isco system fitted with pre-packed silica gel columns (Interchim) or/and preparative HPLC Quaternary Gradient 2545 equipped with a Photodiode Array detector (Waters) fitted with a reverse phase column (XBridge Prep C18 5 μm OBD, 30 × 150 mm). NMR spectroscopy was performed on Bruker spectrometers. Spectra were run in DMSO-d$_6$ or CD$_2$Cl$_2$ or CD$_3$CN, at 298 K. ¹H NMR were recorded at 400–500 MHz, and chemical shifts δ are expressed in ppm using the residual non-deuterated solvent signal as internal standard,

and the coupling constants $J$ are specified in Hz. We only reported labile protons that could be clearly identified in the spectra. [13]C NMR was recorded at 101 or 126 MHz, and chemical shifts δ are expressed in ppm using deuterated solvent signal as internal standard. The purity of final compounds, determined to be >95% by UPLC MS, was recorded on a Waters Acquity H-class equipped with a Photodiode array detector and SQ Detector 2 with a reverse phase column (Aquity UPLC® BEH C18 1.7 μm, 2.1 × 50 mm).

"Classic System": acetonitrile (CAN) (+0.1% FA) and MilliQ Water (+0.1% FA): isocratic at 5% of ACN (0.2 min), then linear gradient from 5% to 100% of ACN in 2.3 min, then isocratic at 100% of ACN (0.5 min). High-resolution mass spectra (HRMS) were recorded on a Thermo Fisher Scientific Q-Exactive Plus equipped with a Robotic TriVersa NanoMate Advion.

Clickable HBH analog HBH-C was synthesized by first preparing compound **1** according to a previously published procedure[74,75] (Supplementary Fig. 7A). To a stirred solution of 4-hydroxy methyl benzoate (304 mg, 2 mmol) and anhydrous $K_2CO_3$ (345 mg, 1.25 eq.) in dry dimethylformamide (DMF) (3.5 mL) at room temperature was added propargylbromide (256 μL, 1.15 eq.). The reaction mixture was stirred at room temperature for 16 h. The mixture solution was extracted with water (20.0 mL) and ethyl acetate (EtOAc) (4 × 15.0 mL). The combined organic phase was dried over sulfate magnesium ($MgSO_4$). The solvent was removed under reduced pressure to afford compound **1** (370 mg, 97%) as a pale brownish solid (Supplementary Fig. 7B, C). Then, HBH-C was prepared according to a previously published procedure[74,75]. Compound **1** (200 mg, 1.05 mmol) and hydrazine hydrate (0.8 mL) were dissolved in methanol (MeOH) (6 mL) at room temperature. The reaction mixture was refluxed overnight. Upon cooling down to room temperature, the resulting mixture was poured into deionized water at 0 °C. The solid was collected and recrystallized from ethanol to afford HBH-C (148 mg, 74%) (Supplementary Fig. 7D–F).

The clickable nifuroxazide analog NIF-C was prepared by reacting nifuroxazide NIF (138 mg, 0.5 mmol) with $Cs_2CO_3$ (179 mg, 1.1 eq.) and propargyl bromide (46 μL, 1.05 eq.) in dry DMF (5 mL) at room temperature overnight. The mixture solution was extracted with water (20.0 mL) and ethyl acetate (EtOAc) (4 × 25.0 mL). The combined organic phases were dried over $MgSO_4$ and concentrated. The crude product was purified by flash column chromatography on silica gel (n-Hex/EtOAC gradient: 0 to 100) to afford the product NIF-C as a yellow powder (Supplementary Fig. 8).

**M2-C** was prepared according to a previously published procedure[76,77]. To 5-nitro-2-furoic acid (314 mg, 2 mmol) in $CH_2Cl_2$, HOBt (1.2 eq.), EDC (1.2 eq.), and propargyl amine (1.05 eq.) were added, and the reaction mixture was stirred for 12 h. After completion of the reaction, water was added. Subsequently, the reaction mixture was washed with saturated $NaHCO_3$ solution followed by addition of 2 N HCl solution. The combined organic layer was separated, dried over anhydrous $Na_2SO_4$, and evaporated on a rotavapor to afford M2-C in good yields (Supplementary Fig. 9).

### Screening strategy
An automated screening routine was developed on a robotic workstation equipped with a 96-well head probe (Nimbus, Hamilton) to screen a repurposing drug library (1280 FDA-approved drugs, Prestwick Chemicals). Briefly, 45 μL of cellular suspension was layered with automation into the wells of collagen-coated, clear bottom, black-walled 384-well culture plates (Greiner μClear plates, Cat# 781091). Starting cell culture conditions were as follows: 1400 MKN45 cells/well; 1800 PANC1 cells/well; 1000 SUM159 cells/well; and 3500 T84 cells/well. Plates were then incubated for 4 h at 37 °C and 5% $CO_2$ in a humidified incubator to allow for cell attachment. Plates were then returned to the robotic workstation, and 5 μL of

sample or control drugs were layered on top of the cell cultures (1 drug/well, final drug concentration: 10 μM). Each drug from the library was tested as a separate triplicate in different well positions of three independent culture plates to minimize positional errors. Each culture plate also received different positive and negative controls: eight wells received medium alone ("Untreated" well, negative controls), twelve received the DMSO vehicle at 0.1% (v/v) final ("DMSO" Wells, negative control, Sigma), four received Doxorubicin at 5 μM final ("Doxo" wells, positive cytotoxic control, Sigma), and four received ST102 at 50 nM final (« ST102 » wells, positive control). Additionally, four wells were left untreated to receive the DEAB control during the ALDEFLUOR assay (see below). Three days post-treatment, cell amount and the %ALDHbr cell amount (=%CSC) were assessed as previously described[23].

### ALDEFLUOR assay
The analysis was processed on single-cell suspension from cell lines. The ALDEFLUOR Kit (Stem Cell Technologies, #01700) was used to isolate population with differential aldehyde-dehydrogenase enzymatic activity and analyzed using an LSR2 cytometer (Becton Dickinson Biosciences) as previously described[4].

### Immunoblot analysis
Cells were lysed in ice-cold lysis buffer containing Hepes 50 nM, pH 7.5, EDTA 1 mM, pH 7, NaCl 150 mM, NaF 100 mM, Na3VO4 1 mM, Triton X-100 1%, and complete Proteinase Inhibitor Cocktail (Roche, #04693159001). Cell lysates were migrated in 4–12% SDS-PAGE (Sodium Dodecyl Sulfate–PolyAcrylamide Gel Electrophoresis). The following primary antibodies were used: anti-ALDH1A1 (mAb, Clone 44, Becton Dickinson, 1/200) anti-CyclinD1 (rabbit mAb, Cell Signaling #55506, 1/1000), anti- p-STAT3 (Tyr705) (mouse mAb, Cell Signaling #4113, 1/2000), anti-STAT3 (mouse mAb, Cell Signaling #9139, 1/1000), anti-FANCD2 (rabbit mAb, abcam, ab108928, 1/1000), anti-FANCI (santa cruz sc-271316, 1/1000) anti-γH2AX (rabbit mAb, Cell Signaling #9718, 1/1000). Detection of GAPDH (Rabbit pAb, Cell Signaling, 1/5000) or α-Actin (mouse mAb, Sigma Aldrich #A5441, 1/5000) was used as loading control.

### Click-chem fluorescence
After cell sorting, cells were cytospun and fixed with 4% paraformaldehyde for 10 min and permeabilized with 0.1% Triton X-100 for 5 min before blocking with protein block (Dako). Click reactions were performed with azide alexa fluor 594 (Click-iT EdU AlexaFluor594 Imaging Kit, C10339) according to the manufacturer's instructions. After 10 min of washing with TBST, DNA was counterstained with DAPI 4',6-diamidino-2-phenylindole (Invitrogen, ProLong Gold antifade reagent with DAPI, P36935). Images were acquired using epifluorescence microscope Leica.

### Immunofluorescence
After cell sorting, cells were cytospun and fixed with 4% paraformaldehyde for 10 min and permeabilized with 0.1% Triton X-100 for 5 min before blocking with protein block (Dako). Cells were labeled 1 h at room temperature with an anti-phospho-Histone H2AX (Ser139, clone JBW301, Merck Millipore, 1/1000) or with an anti-RAD51 (gift from M. Modesti lab, CRCM, Marseille, 1/1000). After 10 min of wash with TBST, cells were incubated for 30 min with anti-mouse (A-11029, ThermoFisher), 1/500. DNA was counterstained with DAPI 4',6-diamidino-2-phenylindole (Invitrogen, ProLong Gold antifade reagent with DAPI, P36935). Images were acquired using Nikon AX confocal microscope equipped with a 63× objective. Cells with more than 8 foci for γH2AX or RAD51 were considered as positive cells. For each condition, immunofluorescence scoring was done on 100 cells in three independent experiments.

## mRNA extraction and quantitative real-time RT-PCR

Total RNA was isolated using the Maxwell RSC simply RNA Tissue Kit (AS1340) according to the manufacturer's instructions. cDNA was synthesized from 1 μg of RNA with the Transcriptase inverse Super-ScriptIV kit (Invitrogen; 18090050). Real-time PCR amplification and analysis were conducted with the TaqMan Universal Master Mix II with UNG on a 7500 Real-Time PCR System (Applied Biosystems). RNA levels were normalized to ACTB expression using the DDCt method. Probe; STAT3 (ThermoFisher, 4331182, Hs00374280_m1) FANCD2 (ThermoFisher, 4331182, Hs00276992_m1) FANCI (ThermoFisher, 4331182, Hs01105308_m1) FANCM (ThermoFisher, 4331182, Hs00326216_m1) FANCA (ThermoFisher, 4331182, Hs01116668_m1) FANCF (ThermoFisher, 4331182, Hs00256030_s1).

## Clonogenic survival analysis

Three hundred SUM159 cells, 500 HeLa cells, 1200 SUM149 cells, and 1500 S68 cells were plated in triplicate in 10 cm dishes in complete growth medium. After cells had attached, they were treated with the indicated dose of compound. After 10–15 days, cells were washed, fixed, and stained with acetic acid: methanol (1:7) and 1% of Coomassie blue 1 h at room temperature. The number of colonies with >100 cells was counted. For each genotype, cell viability of untreated cells was defined as 100%. Data are represented as means ± SD from three independent experiments.

## Reverse comet assay

Reverse alkaline comet assay was done as described previously[78]. Briefly, cells were initially treated with DMSO vehicle or NIF. After 4 h of treatment and cell sorting, cell samples were treated with PBS as a control or IR (10 Gy). Cells were subsequently resuspended in molten 1% Type VII low gelling temperature agarose and then allowed to set on homemade glass slides pre-coated with agarose[78]. Cells were then lysed by bathing slides in ice-cold lysis buffer (100 mM Na2EDTA, 2.5 M NaCl, 10 mM Tris–HCl (pH 10.5), 1% Triton X-100) for 60 min and then subjected to 4 × 15 min washes with ice-cold MilliQ H2O. Each slide was then submerged in alkali electrophoresis buffer (300 mM NaOH, 1 mM Na2EDTA) for 60 min and then electrophoresed at 20 V for 20 min at 4 °C. Samples were neutralized by the addition of neutralization buffer (500 mM Tris–HCl pH 7.5) for 10 min and then allowed to dry overnight at ambient temperature. Comets were stained with SYBR Green for 10 min and then washed using 3× MilliQ H2O washes. Samples were visualized using an Apotome microscope, and the level of DNA damage was assessed using OpenComet. At least 100 comets were scored per slide.

## Quantitative image-based cytometry (QIBC)

The nucleus was extracted and stained as previously described[36]. Briefly, after treatment and 45 min of EdU incubation, the nucleus was extracted with CSK buffer (50 mM NaCl, 25 mM Hepes pH7.4, 3 mM MgCl2, 300 mM sucrose, 0.5% triton, 1 mM EDTA and protease and phosphatase inhibitor) 5 min on ice. The buffer was removed, and the nucleus was fixed with PFA 2% 10 min at room temperature. For click analysis, the nucleus was stained with azide alexaFluor 594 (Click-iT EdU AlexaFluor594 Imaging Kit, C10339) according to the manufacturer's instructions, washed nucleus with Washing buffer (BD Perm/Wash, 51-2091KZ). The nucleus was stained with analysis buffer (DAPI 0.5 μg/ml and RNase 250 μg/ml). For cell cycle analysis, nuclei were stained only with an analysis buffer. These nuclei are analyzed by cytometry on Aurora (Cytek).

## Tumosphere forming assay

Cell lines were plated on 96-well plates (pre-coated with Poly(2-hydroxyethyl methacrylate) at 56 °C overnight) in serum-free mammary epithelial basal medium (MEBM, Lonza, CC-3151) supplemented with B-27 (Gibco, 17504-004), 20 ng/mL EGF (Gibco, PMG8043), 1X Antibiotic-Antimycotic (Gibco, 15240-062), 1 ng/mL hydrocortisone (Sigma, H0888), 5 μg/mL insulin (Lily, VL7510, from IPC). Frequency of tumorigenic cells with tumorsphere-forming ability was determined following the guidelines of the Extreme Limiting Dilution Analysis (ELDA)[79]. Briefly, a range of cells (1–25 for SUM159 and 1–100 for S68 and SUM149) was plated in each well, and the number of wells containing at least one tumorsphere was computed after 10–15 days of culture. 30–80 wells were evaluated per condition.

## Apoptosis assay

After cell sorting, SUM159 cell line was treated in adherent conditions for 72 h. According to the manufacturer's instructions, at the end of treatment, the cells were resuspended in binding buffer 1X (0.1 M hepes pH = 7.4, 1.4 M NaCl, and 25 mM CaCl2 for 10X buffer), with 100,000 cells in 100 μl of buffer. Then, cells were stained with 5 μl of Annexin V-FITC (ThermoFisher; A13201) and 10 μl of Propidium Iodide (ThermoFisher; 556463). Cells were incubated 15 min at RT in dark and analyzed by spectral flow cytometry.

## RNAseq

For analysis of SUM159 and SW620, three independent experiments of ALDHneg and ALDHbr cells were isolated by cell sorting for control condition of treated by NIF during 30 h. Total RNA was extracted as described above, and its quality was assessed by Tapestation (only samples with RIN score > 8 were considered for sequencing). The sequencing and GSEA analysis were performed by MGX-Montpellier GenomiX core facility.

## CUT&RUN qPCR

Cut&Run was performed as described previously[80]. Briefly, after 1 or 6 h of IL-6 stimulation, SUM159 cells were washed twice with Wash buffer (20 mM HEPES pH = 7.5, 150 mM NaCl, 0.5 mM spermidine (Sigma; S2626-5G) supplemented with protease inhibitors). Cells were then resuspended in Binding buffer (20 mM HEPES pH = 7.9, 10 mM KCl, 1 mM CaCl2, 1 mM MnCl2) containing 10 μL of blocked BioMag®-Plus Concanavalin A-coated beads (CliniSciences, #86057-10). After 10 min at RT, cells:beads were transferred to 50 μL of antibody solution (1:100 dilution of primary antibody (STAT3; ab171360 and H3K27ac; active motif; #39133) in Wash buffer plus 0.1% digitonin) and incubated for 1 h at RT on an end-to-end rotator. Cells:beads were washed three times with Wash buffer-0.1% digitonin and incubated for 10 min with pA-MNase (given by E. Pasquier, CRCM, France)[80] diluted in Wash buffer-0.1% digitonin. After three washes in Wash buffer-0.1% digitonin, tubes were placed in an ice/water bath and equilibrated to 0 °C for 10 min. To trigger digestion of the DNA, CaCl2 was added (2 mM final concentration) for 30 min. Digestion was stopped by addition of 2X STOP buffer (340 mM NaCl, 20 mM EDTA, 4 mM EGTA, 0.02% digitonin, 1:200 RNase A [50 μg/mL final concentration]). Digested DNA fragments were released from cells by incubating sample tubes on a heat block at 37 °C for 10 min. The supernatant was then recovered by placing tubes on a magnetic stand, and DNA was purified using the NucleoSpin Gel and PCR Clean-up kit (#28104). DNA fragment has been analyzed by q-PCR in 384 wells with iQ SYBRE green supermix according to the manufacturer's instructions. We used the probes for STAT3 promoter (Fw: ATGACCGGAATGTCCTGCTG and Rv: TCACGCACTGCCAGGAAC) FANCI (Fw: CTCCGACTGTGAGCTGGGA and Rv: ATGAAGACTGAAGGGGTGCC). Data were normalized with GAPDH probe (Fw: ACTCACCCTGCCCTCAATATC and Rv: AGACAG TGTGCCTTTCATTCCAT).

## FANCI reporter

To study effect of STAT3 and M2 on FANCI expression, we used a reporter assay (LightSwitch Luciferase vector) with luciferase expression under the control of FANCI or GAPDH promoter as control vector (Active Motif). Twenty-four hours post-transfection, luciferase

expression was revealed using a LightSwitch Dual assay kit (Active Motif; #32031) according to the manufacturer's instructions. Data were normalized to the GAPDH expression.

## Mutation and copy-number detection

For each PDX, we identified molecular alterations array-comparative genomic hybridization (aCGH) as previously described[60]. aCGH was done using high-resolution 244 K CGH microarrays (Hu-244A, Agilent Technologies). To determine copy-number alterations in each PDX, we mapped all aCGH probes according to the hg19/NCBI human genome mapping database. The copy number was estimated for each gene by taking the value of the segment with the highest amplitude, then categorized into "Amplified" (Log2ratio > 1), "Gain" (0.5 < Log2ratio ≤ 1), "Loss" (−1 ≤ Log2ratio < −0.3), and "deletion" (Log2ratio < −1). Focal events were defined as genomic alterations with a size less than 5 Mb and a copy number higher than the surrounding segments. The percentage of genome altered was calculated as the sum of altered probes divided by the total number of probes. To determine the mutation profile of each PDX, we combined two data analysis pipelines as described previously[60]. Briefly, the first pipeline used FreeBayes version 0.9.9 for single-nucleotide variant (SNV) calling, and insertions/deletion (indel) calling was done using GATK haplotype caller version 2.5-gf57256b with default parameters. For the second pipeline, SNV calling was done with Mutect 1.7 and somatic indel calling with scalpel. All variants were then annotated for genes and function using ANNOVAR (version 2013-1112). In order to remove false positives, recurrent variants with no entry in public databases such as COSMIC or dbsnp were removed. Variants identified by both pipeline analyzes were retained as somatic.

## Animal models

In this study, we utilized four primary human breast cancer xenografts generated from four different patients (CRCM434, CRCM494, Pandora21, and Pandora7). These patient-derived xenografts (PDXs) were generated triple-negative breast tumors[59]. For each PDX, we determine HRD status using the SOPHiA DDM™ GIInger Genomic Integrity Solution that is based on Low-pass whole genome sequencing. We utilized these PDXs to perform preclinical assay in vivo. Cells from these PDXs were transplanted orthotopically into fat pads of NSG female mice that were between 6 and 10 weeks old without cultivation in vitro. We injected 100,000 cells per fat pad of NSG mice (with one injected fat pad per mouse) and monitored tumor growth. When tumors reached an average size of 10–150 mm³, mice were randomized (n = 8, i.e., 8 tumors for each PDX and for each group) and used to determine the response to the treatment. We initiated treatment with NIF (i.p., 300 mg/kg, 5 out of 7 days, 3 weeks), alone, PARPi alone (i.p., 50 mg/kg, 5 out of 7 days, 3–4 weeks), NIF/PARPi combination, or placebo injected with a solution of 12.5% ethanol/12.5% cremophore/75% water. After 3–4 weeks of treatment, mice from each group were sacrificed according to ethical statements. Of note, in this protocol, no mouse was excluded for ethical issues due to adverse treatment issue. Tumors were dissociated into single cells. Live cells were counted using the LUNA-FL™ Dual Fluorescence Cell Counter (Logos biosystems) and reimplanted into secondary NSG mice. We performed serial dilution to functionally evaluate the proportion of residual CSCs in each group of treatment (CTRL, NIF, PARPi, and NIF + PARPi) from the 4 different PDXs. Each mouse that presented a tumor reaching a size of 100 mm³ was considered as a tumor-bearing mouse.

## Patient-derived organoids (PDXOs)

To grow organoids from PDX models (CRCM494, CRCM434, Pandora21, and Pandora7), 250,000 cells were resuspended in 28 μL of culturex (Biotechne), seeded on a 48-well plate, and cultured in 400 μL of medium supplemented with 10 μM of L-Y27632 (Sigma, G9145) as previously described[81]. After 7–10 culture days, PDXOs were passed,

and only PDXOs > 40 μm were replated (400 organoids per well) in 400 μL of medium without L-Y27632. After 3–7 days of culture, PDXOs were treated with 25 μM of M2-C or Melphalan-C for 4 h. After treatment, all wells were washed twice with PBS, and 400 μL of fresh medium was added. After 02,472 and 144 h of treatment, PDXOs were dissociated with TrypLE Express 1× (Gibco, #12605-010) during 15 min at 37 °C under agitation (155 RPM), fixed in 70% ethanol, and immediately stored at −20 °C. For ICLick analysis, cells were stained with azide Alexa Fluor 594 (Click-iT EdU AlexaFluor594 Imaging Kit, C10339) according to the manufacturer's instructions. Cells were stained with analysis buffer (DAPI 1/5000). These cells were analyzed by cytometry on LSRII (BD).

## Statistics and reproducibility

Graphpad Prism 5.0 was used for data analysis and imaging. The results are presented as mean ± SD for at least three repeated independent experiments. To investigate associations among variables, using nonparametric Wilcoxon rank-sum test, ANOVA and Sidak or Dunn's test, one-sided chi-squared test or one-sided Fisher's exact test when appropriate. Extreme limiting-dilution analysis (http://bioinf.wehi.edu.au/software/elda/) was used to evaluate breast CSC frequency. In all cases, a p value < 0.05 was considered statistically significant. No statistical method was used to predetermine sample size. No data were excluded from the analyzes. The experiments were not randomized, and the investigators were not blinded to allocation during experiments and outcome assessment.

## Reporting summary

Further information on research design is available in the Nature Portfolio Reporting Summary linked to this article.

## Data availability

RNAseq dataset generated for this publication is deposited in the Gene Expression Omnibus (GEO) repository (https://www-ncbi-nlm-nih-gov/geo/) under the accession number GSE288275. All the other data are available in the article, Supplementary information file, or source data file. Source data are provided with this paper.

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

## Acknowledgements
We thank Raphael Ceccaldi and Jean-Hugues Guervilly for their valuable comments on the manuscript. Work in the lab of C.G., J.P., and R.R. was supported by a grant from Plan Cancer (2014-2019). C.G. is supported by Inserm, Institut-Paoli-Calmettes, Canceropole PACA, Fondation ARC, and the "Ligue National Contre le Cancer" (Equipe Labellisée). R.R. is funded by the European Research Council (grant agreement No 647973), Ligue Contre le Cancer (Equipe Labellisée), Fondation Charles Defforey-Institut de France, Klaus Grohe Foundation, and Foundation Bettencourt Schueller. C.D.R. has been supported by a fellowship from the "Ligue National Contre le Cancer" and L.B. by a fellowship from CNRS (MITI Prime 22). We thank the Canceropôle PACA, IBISA, and the Plan Cancer Equipement (#17CQ047-00) for continued support in the development of the TrGET preclinical assay platform (CRCM). We thank Gisele Froment, Didier Negre, and Caroline Costa from the lentivectors production at facility/SFR BioSciences Gerland-Lyon Sud (UMS3444/US8). Thanks are due to the MGX-Montpellier GenomiX, MRI flow cytometry platforms, CRCM flow cytometry and Imaging platform, the 3D-HubO platform, and CRCM animal core facilities.

## Author contributions
C.D.R., R.R., J.P., J-M.P., C.L., E.C.-J., and C.G. conceived the study and designed the experiments with input from all authors. C.D.R., L.B., E.M.B., S.L., and M.M. performed most experiments. C.B. and R.C. performed the in vivo experiments. S.D. and L.C. designed and generated all click-derivatives. J.W., V.C., and G.G. performed organoid cultures. G.P. and M.V. designed and performed the drug screen. O.R. designed and generated the CRISPRi cell lines. C.P. and H.S. performed the HRD score. E.P. provided technical support. E.C.J. & C.G. supervised the work with input from C.L. and his lab. C.D.R., C.L., E.C.J., and C.G. prepared the manuscript with input from all authors.

## Competing interests
The authors declare no competing interest
