## [Transparent Peer Review file · Nature Communications]

Inhibition of the STAT3/Fanconi anemia axis is synthetic lethal with PARP inhibition in breast cancer

Corresponding Author: Dr Christophe Ginestier

Version 0:

Reviewer comments:

Reviewer #1

(Remarks to the Author)

The authors screened a repurposing library to find compounds which inhibit ALDH activity, one of the properties in cancer stem cells and found Nifuroxazide (NIF), an antibiotics, having such activity. They found that ALDH catalyzes NIF to generate a cytotoxic metabolite M2. M2 appears to generate DNA interstrand crosslink and inhibits the FA pathway responsible of the DNA interstrand crosslink repair, leading to cell death in ALDH-high cells. STAT3 might bind to the promoters of FA pathway genes and induce their transcription for DNA interstrand crosslink repair, leading to cell survival. M2 inhibits STAT3 activity. These dual activities of M2 confer vulnerability in ALDH-high cells. Then they showed that NIF treatment sensitizes homologous recombination proficient (HRP) cells against PARPi in vitro. Finally, they showed that combinatory treatment of NIF and PARPi may reduce number of CSCs in PDX model.

It is reported that NIF inhibits STAT3 in breast cancer (Wang X, Cell Death Discov, 2023) and that NIF eradicates ALDH-high melanoma-initiating cells through inhibition of ALDH activity (Sarvi S, Cell Chem Biol, 2018). Although in this manuscript they showed that NIF generates DNA interstrand crosslink and increases sensitivity against PARPi in ALDH-high HRP cells in vitro, the results of PDX models derived from HRP cancer tissues did not show consistent findings. It is hard to conclude NIF induces a chemical HRDness that is synthetic lethal with PARPi.

Reviewer #2

(Remarks to the Author)

Summary: This paper finds that the antibiotic NIF is specifically activated in breast cancer stem cells, where it simultaneously induces DNA ICLs and suppresses FA pathway transcription through inhibition of STAT3, thereby inducing synthetic lethality with PARP inhibition. These findings have significant translational potential, as NIF could be used to selectively sensitize breast cancer stem cells to PARP inhibitors.

Abstract

-line 51, please define homologous recombination deficiency for abbreviation (HRDness)

-Introduction, line 68, please define epithelial-to-mesenchymal transition abbreviation (EMT)

Results

-Fig 1D, is the tumorsphere formation of PANC1 significantly reduced by NIF? No significance is indicated on the graph, but the text states that there was "reduction of the tumorsphere-forming efficiency in all NIF-treated cell lines" (line 140-141). Please clarify.

-line 229-230, suggest clarification that gamma-H2AX marks DNA double-strand breaks, which can result from replication stress. If other literature supports gamma-H2AX as a marker of replication stress in the absence of resultant DSBs, please reference.

-Sup. Fig 3B, please specify length of STAT3i treatment. If STAT3 inhibition suppresses FA pathway-mediated repair of DNA breaks, you would expect to see an accumulation of gamma-H2AX over time.

-line 252, It would be helpful to readers to clarify that both FANCD2 and ERCC1 are members of the FA pathway.

-line 379, please clarify that BRCA1/2 are part of the FA pathway.

Overall, the authors did an excellent job of utilizing appropriate methodology to address their questions, which were prompted by clear rationale. They provide robust data that supports their conclusions. Their findings offer high translational

value for improved breast cancer treatment strategies. I note only minor concerns with potentially confusing language, which can be easily addressed with editorial assistance.

Reviewer #3

(Remarks to the Author)

Reviewer #4

(Remarks to the Author)

This is an interesting paper that gives more insight in the sensitivity of cancer stem/progenitor cells to Nifuroxazide. The authors confirm previously reported metabolism of Nifuroxazide in ALDH^{High} cancer stem/progenitor cells and show that the M2 metabolite causes DNA ICLs and also decreased expression of FA-related genes through inhibition of STAT3. This is associated with an HR defect that sensitizes HR proficient breast cancer cells to PARPi. The authors use several cleverly designed assays and the data appear generally robust. However, there are some issues that need to be addressed.

Major comments

1. Using the same Prestwick drug repurposing library, Nifuroxazide has already been identified before as a potent STAT3 inhibitor (Nelson et al., Blood 2008, DOI: 10.1182/blood-2007-12-129718). This paper is cited (ref 30), but the authors should also mention the striking overlap between the results of the STAT3 inhibitor screen and their Aldefluor-based cancer stem cell screen performed with the same drug library. This will allow the reader to get a better understanding of the value of the drug library and the mechanisms underlying the results of the screens.
2. It is unclear why STAT3 depletion alone can mimic the effect of Nifuroxazide on tumorsphere-formation in the SW620 colon cancer cell line, whereas the breast cancer cell lines tested are not affected. Is this because of the higher knockdown efficiency in SW620 cells or is there a difference in the levels of ALDH1 expression and/or ICL repair capacity between these cell lines? Inhibitors affecting STAT3 signaling were only tested on the breast cancer cell lines and it is not clear if they have been used at effective doses. To clarify if different STAT3 knockdown efficiencies may underlie differences in tumorsphere formation, the authors should either achieve comparable knockdown or confirm the knockdown data for SW620 with inhibitors. For the latter, it is also necessary to use and mention effective dose levels.
3. Conversely, it is unclear why the authors conclude in their discussion (lines 492-494) that "contrary to what has been reported in melanomas, ALDH inhibition is not sufficient to explain the anti-CSC effect of NIF". As shown in Figure 1G, knockdown of ALDH1A1 abrogates the inhibiting effect of Nifuroxazide on breast cancer cell line sphere formation. This does not necessarily mean that ALDH inhibition is sufficient to explain the anti-CSC effect of Nifuroxazide, but it also does not show that other factors are required. To get more insight in the dependence of these breast cancer cells on ALDH1A1 for ALDH activity, Aldefluor assays should be done. Moreover, treatment of ALDH1A1 knockdown cells with the Nifuroxazide metabolite M2 would show if sphere formation is affected by factors beyond ALDH inhibition.
4. Throughout the manuscript (e.g., line 161-162), the phrasing suggests that results can be extrapolated to 'breast cancers' in general. This would be an oversimplification, as breast cancers have very different molecular characteristics. In the absence of an explanation of sensitivity to Nifuroxazide-mediated STAT3/FA axis inhibition, the authors should at least make clear that their observations cannot be extrapolated to breast cancer vs other cancers in general. In addition, they only used triple negative breast cancer models, this should be mentioned explicitly.
5. The in vivo experiments need some clarification, with the current description I cannot evaluate the validity of all conclusions. Did the authors take out any animals due to adverse treatment effects? How many mice were taken along for the data presented in the graphs? (needs specification in all (SI) graphs and/or (SI) Figure legends). The in vivo ELDA experiment description mentions that 5-100 cells were transplanted. Were differences in viability (upon treatment) taken into account? Procedures need to be described in more detail and the Kaplan-Meier plots of SI Figure 6 lack clarity (e.g., SI Figure 6E seems to have some partially overlapping curves and the Mantel-Cox p-values cannot be understood based on the graph).
6. In the Discussion, it should be acknowledged that systemic chemical HRDness may increase side effects of PARP inhibition.

Minor comments

1. The text contains some small errors, please check for missing words and characters. Not sure if the title should state 'the STAT3/FA axis' instead of 'STAT3/FA axis' and if it should be 'PARP inhibition' or 'the PARP inhibitor olaparib', instead of 'PARP inhibitor'. In the Methods, the heading 'Statistical' should be 'Statistics' or something like 'Statistical Analyses'.

Version 1:

Reviewer comments:

Reviewer #1

(Remarks to the Author)

They conducted additional experiments using the HRP PDX model and now present convincing data. The reviewer is

satisfied with their findings.

Reviewer #3

(Remarks to the Author)

Reviewer #4

(Remarks to the Author)

The authors have adequately addressed my concerns, I have no further comments.

Summary: This paper finds that the antibiotic NIF is specifically activated in breast cancer stem cells, where it simultaneously induces DNA ICLs and suppresses FA pathway transcription through inhibition of STAT3, thereby inducing synthetic lethality with PARP inhibition. These findings have significant translational potential, as NIF could be used to selectively sensitize breast cancer stem cells to PARP inhibitors.

Abstract

-line 51, please define homologous recombination deficiency for abbreviation (HRDness)

-Introduction, line 68, please define epithelial-to-mesenchymal transition abbreviation (EMT)

Results

-Fig 1D, is the tumorsphere formation of PANC1 significantly reduced by NIF? No significance is indicated on the graph, but the text states that there was “reduction of the tumorsphere-forming efficiency in all NIF-treated cell lines” (line 140-141). Please clarify.

-line 229-230, suggest clarification that gamma-H2AX marks DNA double-strand breaks, which can result from replication stress. If other literature supports gamma-H2AX as a marker of replication stress in the absence of resultant DSBs, please reference.

-Sup. Fig 3B, please specify length of STAT3i treatment. If STAT3 inhibition suppresses FA pathway-mediated repair of DNA breaks, you would expect to see an accumulation of gamma-H2AX over time.

-line 252, It would be helpful to readers to clarify that both FANCD2 and ERCC1 are members of the FA pathway.

-line 379, please clarify that BRCA1/2 are part of the FA pathway.

Overall, the authors did an excellent job of utilizing appropriate methodology to address their questions, which were prompted by clear rationale. They provide robust data that supports their conclusions. Their findings offer high translational value for improved breast cancer treatment strategies. I note only minor concerns with potentially confusing language, which can be easily addressed with editorial assistance.

First, we would like to thank the reviewers for their time and constructive feedbacks. We provide here a point-by-point response to all the reviewers' comments.

Reviewer #1 (anti-CSC cancer therapy):

The authors screened a repurposing library to find compounds which inhibit ALDH activity, one of the properties in cancer stem cells and found Nifuroxazide (NIF), an antibiotics, having such activity. They found that ALDH catalyzes NIF to generate a cytotoxic metabolite M2. M2 appears to generate DNA interstrand crosslink and inhibits the FA pathway responsible of the DNA interstrand crosslink repair, leading to cell death in ALDH-high cells. STAT3 might bind to the promoters of FA pathway genes and induce their transcription for DNA interstrand crosslink repair, leading to cell survival. M2 inhibits STAT3 activity. These dual activities of M2 confer vulnerability in ALDH-high cells. Then they showed that NIF treatment sensitizes homologous recombination proficient (HRP) cells against PARPi in vitro. Finally, they showed that combinatory treatment of NIF and PARPi may reduce number of CSCs in PDX model.

It is reported that NIF inhibits STAT3 in breast cancer (Wang X, Cell Death Discov, 2023) and that NIF eradicates ALDH-high melanoma-initiating cells through inhibition of ALDH activity (Sarvi S, Cell Chem Biol, 2018). Although in this manuscript they showed that NIF generates DNA interstrand crosslink and increases sensitivity against PARPi in ALDH-high HRP cells in vitro, the results of PDX models derived from HRP cancer tissues did not show consistent findings. It is hard to conclude NIF induces a chemical HRDness that is synthetic lethal with PARPi.

Knowing that despite initial responsiveness to PARPi, HRD tumors eventually develop resistance through a variety of mechanisms, it is crucial to enhance PARPi efficacy in both acquired and also potentially innately (HRP tumors) PARPi-resistant tumors. In this manuscript we report that NIF is synthetic lethal with PARPi in both type of PARPi-resistant tumors. We used two different PDX models of PARPi-resistant tumors initially predicted HRD by genomic integrity index (CRCM494 and Pandora21). In both models, we obtain similar results with residual cells isolated from PARPi-treated tumors in combination with NIF that presented a drastic reduction in the tumor-initiating capacity in secondary mice compared with control, PARPi, or NIF alone (**Figure 6 G-H, Supplementary Figure 6 C-D**).

Nonetheless, we agree with the reviewer that our result obtain in the HRP PDX model (Pandora 7) was less convincing. We suspected that the number of cells used in the reimplantation assay was not adapted to the high tumorigenic potential of this model. We have now repeated this experiment by xenotransplanting residual cells isolated from PARPi-treated tumors in combination with NIF or treated in monotherapy (NIF or PARPi alone) compared to the control using a new range of serial dilution (50, 25, 10, and 5 cells; for at least 3 replicates per dilution). By combining these new in vivo experiments with previous ones, we were able to demonstrate that residual cells isolated from PARPi-treated tumors in combination of NIF display a significant reduction in the tumor-initiating capacity in secondary mice compared to the control. All these results are now reported in **Figure 6 I and Supplementary Figure 6 E**.

Reviewer #2 (Fanconi anaemia therapy/signaling):

Summary: This paper finds that the antibiotic NIF is specifically activated in breast cancer stem cells, where it simultaneously induces DNA ICLs and suppresses FA pathway transcription through inhibition of STAT3, thereby inducing synthetic lethality with PARP inhibition. These findings have significant translational potential, as NIF could be used to selectively sensitize breast cancer stem cells to PARP

inhibitors.

Abstract

-line 51, please define homologous recombination deficiency for abbreviation (HRDness)

We made the correction

-Introduction, line 68, please define epithelial-to-mesenchymal transition abbreviation (EMT)

We made the correction

Results

-Fig 1D, is the tumorsphere formation of PANC1 significantly reduced by NIF? No significance is indicated on the graph, but the text states that there was "reduction of the tumorsphere-forming efficiency in all NIF-treated cell lines" (line 140-141). Please clarify.

We thank the reviewer for pointing out this oversight and we made the correction by indicating significance on the graph (**Figure 1D**)

-line 229-230, suggest clarification that gamma-H2AX marks DNA double-strand breaks, which can result from replication stress. If other literature supports gamma-H2AX as a marker of replication stress in the absence of resultant DSBs, please reference.

We have now cited a study that demonstrate that in addition to its role in the recognition and repair of double strand breaks, H2AX also participates in the surveillance of DNA replication (Ward and Chen, JBC, 2001)

-Sup. Fig 3B, please specify length of STAT3i treatment. If STAT3 inhibition suppresses FA pathway-mediated repair of DNA breaks, you would expect to see an accumulation of gamma-H2AX over time.

We have now specified in the figure legend that cells were treated with STAT3i for 3 days. Additionally, following Reviewer 2's hypothesis, we exposed cells to STAT3i over an extended period (10 days of treatment) and evaluated DNA breaks by measuring gH2AX levels. As shown in the figure below, we did not detect any significant accumulation of gH2AX over time. This observation may be explained by the low level of spontaneous DNA damaged formed in these cells, which therefore do not require FA pathway activity. This hypothesis is supported by the low level of basal FANCD2 and FANCI mono-ubiquitination in these cells, as reported in Figure 4A. These observations further highlight the therapeutic potential of using NIF, a dual-acting drug that first induces DNA ICLs and then inhibits the FA pathway responsible for ICL repair.

-line 252, It would be helpful to readers to clarify that both FANCD2 and ERCC1 are members of the FA pathway.

-line 379, please clarify that BRCA1/2 are part of the FA pathway.

We made the clarification in the text and add a reference describing the FA pathway players (Ceccaldi et al., Nat Rev Mol Cell Biol, 2016).

Overall, the authors did an excellent job of utilizing appropriate methodology to address their questions, which were prompted by clear rationale. They provide robust data that supports their conclusions. Their findings offer high translational value for improved breast cancer treatment strategies. I note only minor concerns with potentially confusing language, which can be easily addressed with editorial assistance.

Reviewer #3 (co-reviewer):

Reviewer #4 (DNA repair, cancer therapy, CSC):

This is an interesting paper that gives more insight in the sensitivity of cancer stem/progenitor cells to Nifuroxazide. The authors confirm previously reported metabolism of Nifuroxazide in ALDH^{High} cancer stem/progenitor cells and show that the M2 metabolite causes DNA ICLs and also decreased expression of FA-related genes through inhibition of STAT3. This is associated with an HR defect that sensitizes HR proficient breast cancer cells to PARPi. The authors use several cleverly designed assays and the data appear generally robust. However, there are some issues that need to be addressed.

Major comments

1. Using the same Prestwick drug repurposing library, Nifuroxazide has already been identified before as a potent STAT3 inhibitor (Nelson et al., Blood 2008, DOI: 10.1182/blood-2007-12-129718). This paper is cited (ref 30), but the authors should also mention the striking overlap between the results of the STAT3 inhibitor screen and their Aldefluor-based cancer stem cell screen performed with the same drug library. This will allow the reader to get a better understanding of the value of the drug library and the mechanisms underlying the results of the screens.

We agree with the reviewer that the identification of Nifuroxazide as an inhibitor of tumorigenic cell survival from various pathologies, achieved through two independent screens of the same drug library, further supports the potential of this screening approach to discover new anti-cancer drug candidates. Furthermore, the overlap between drug repurposing studies presents an opportunity to elucidate underlying anti-cancer mechanisms. To strengthen this point we have now modified the text as follow:

“Of note, NIF has been identified as a potent anti-multiple myeloma drug by screening the PRESTWICK repurposing library³⁰, thereby presenting an opportunity to elucidate underlying anti-cancer mechanisms. In this previous study, NIF was described as a potent STAT3 inhibitor. Therefore, we hypothesized that the anti-CSC effect of NIF might be mediated through this actionable target.”

2. It is unclear why STAT3 depletion alone can mimic the effect of Nifuroxazide on tumorsphere-formation in the SW620 colon cancer cell line, whereas the breast cancer cell lines tested are not

affected. Is this because of the higher knockdown efficiency in SW620 cells or is there a difference in the levels of ALDH1 expression and/or ICL repair capacity between these cell lines? Inhibitors affecting STAT3 signaling were only tested on the breast cancer cell lines and it is not clear if they have been used at effective doses. To clarify if different STAT3 knockdown efficiencies may underlie differences in tumorsphere formation, the authors should either achieve comparable knockdown or confirm the knockdown data for SW620 with inhibitors. For the latter, it is also necessary to use and mention effective dose levels.

The reviewer raised a crucial question: How can we account for the differences observed between colon cancer and breast cancer regarding the impact of STAT3 depletion on the cancer stem cell (CSC) population? Are these differences due to technical issues or genuine biological effects?

To rule out any technical issues, we first validated that similar knockdown (KD) efficiencies of STAT3 expression were achieved in both breast and colon cells (e.g., SUM159-KRAB sgSTAT3#1 vs. SW620-KRAB sgSTAT3#1) (**Supplementary Figure 1E**). Despite this, we observed a decrease in sphere-forming efficiency (SFE) in colon cells, but not in breast cells (**Supplementary Figure 1G, I**). Furthermore, we now measured ALDH activity in SW620-KRAB cells following STAT3 KD and found a significant reduction in the ALDH^{br} cell population that was not observed in breast cells (**new Supplementary Figure 1F**).

Following the reviewer's suggestion, we tested the effect of inhibitors targeting STAT3 signaling on the ALDH^{br} cell proportion and SFE in SW620 cells. Only the STAT3 inhibitor (Napabucasin), and not the JAK2 inhibitor (Ruxolitinib), efficiently decreased STAT3 activation, as shown by the reduced pSTAT3/STAT3 ratio (**new Supplementary Figure 1J**). This reduction was comparable to that observed in breast cancer cells treated with STAT3 or JAK2 inhibitors. Moreover, Napabucasin significantly reduced the ALDH^{br} cell population and SFE of treated SW620 cells (**new Supplementary Figure 1K, L**), while no impact was observed in breast cell lines.

These additional observations further confirm that STAT3 inhibition is sufficient to reduce the CSC population in colon cancers, but not in breast cancers, and that this is not due to differing STAT3 KD efficiencies. Of note, STAT3 signaling is known to be constitutively active in colon CSC and Napabucasin has been described as a first-in-class cancer stemness inhibitor in colorectal cancer, with a phase 3 trial demonstrating longer overall survival in Napabucasin-treated patients with pSTAT3-positive tumors (Jonker et al., *Lancet Gastroenterol Hepatol*, 2018).

Overall, these observations further demonstrate that colon CSC survival is dependent on STAT3 activation, unlike breast CSCs

Concerning dose concentrations used for each drug to treat each cell model we have now added a **supplementary table 1** reporting each treatment condition.

3. Conversely, it is unclear why the authors conclude in their discussion (lines 492-494) that “contrary to what has been reported in melanomas, ALDH inhibition is not sufficient to explain the anti-CSC effect of NIF”. As shown in Figure 1G, knockdown of ALDH1A1 abrogates the inhibiting effect of Nifuroxazide on breast cancer cell line sphere formation. This does not necessarily mean that ALDH inhibition is sufficient to explain the anti-CSC effect of Nifuroxazide, but it also does not show that other factors are required. To get more insight in the dependence of these breast cancer cells on ALDH1A1 for ALDH activity, Aldefluor assays should be done. Moreover, treatment of ALDH1A1 knockdown cells with the Nifuroxazide metabolite M2 would show if sphere formation is affected by factors beyond ALDH inhibition.

We agree with the reviewer that this point was confusing. Our intention was to say that ALDH1A1 depletion “alone” is insufficient to explain the anti-CSC effect of NIF, as shown in **Figure 1G** (CTRL-sgEmpty vs. CTRL-sgALDH). However, we fully agree that ALDH1A1 enzymatic activity is indeed essential in mediating the anti-CSC effect of NIF (as shown in **Figure 1G**; NIF-sgEmpty vs. NIF-sgALDH).

To avoid confusion, we modified the text as follows: "*ALDH inhibition alone is insufficient to explain the anti-CSC effect of NIF.*"

Following the reviewer's suggestion, we assessed ALDH activity after ALDH1A1 knockdown and confirmed that ALDH activity in SUM159 and S68 cells predominantly depends on ALDH1A1 expression (**new Supplementary Figure 1Q**). Additionally, we treated wild-type (WT) and ALDH1A1 KD cells with the metabolite M2 and conducted a tumorsphere assay. M2 treatment reduced sphere-forming efficiency (SFE) similarly in both WT and ALDH1A1 KD cells, further demonstrating that while ALDH activity is required for NIF bioactivation, the anti-CSC effect of NIF and its bioactivated metabolite is beyond ALDH inhibition. These results are now reported in (**Figure 1J**).

4. Throughout the manuscript (e.g., line 161-162), the phrasing suggests that results can be extrapolated to 'breast cancers' in general. This would be an oversimplification, as breast cancers have very different molecular characteristics. In the absence of an explanation of sensitivity to Nifuroxazide-mediated STAT3/FA axis inhibition, the authors should at least make clear that their observations cannot be extrapolated to breast cancer vs other cancers in general. In addition, they only used triple negative breast cancer models, this should be mentioned explicitly.

We apologize for any confusion experienced by the reviewer. We did not mean to assert that our findings should be generalized to all breast cancers, but rather point out that our in vitro experiments were performed using cell lines of different molecular subtypes (a basal/mesenchymal cell line and a luminal cell line). Therefore, as indicated in the discussion, our results provide a rationale for further investigation using a larger panel of models, including other cancers treated with PARP inhibitors (PARPi), such as ovarian cancer.

For the in vivo experiments, we deliberately chose patient-derived triple-negative breast cancer models because they are likely to receive PARPi treatment. Although the molecular characteristics of our models have already been mentioned in the Materials and Methods section, we have now explicitly stated this information in the Results section for greater clarity.

5. The in vivo experiments need some clarification, with the current description I cannot evaluate the validity of all conclusions. Did the authors take out any animals due to adverse treatment effects? How many mice were taken along for the data presented in the graphs? (needs specification in all (SI) graphs and/or (SI) Figure legends). The in vivo ELDA experiment description mentions that 5-100 cells were transplanted. Were differences in viability (upon treatment) taken into account? Procedures need to be described in more detail and the Kaplan-Meier plots of SI Figure 6 lack clarity (e.g., SI Figure 6E seems to have some partially overlapping curves and the Mantel-Cox p-values cannot be understood based on the graph).

We have made clarifications in the Material and Methods section to better describe our procedure. We specify that in this protocol, no mice were excluded for ethical reasons due to adverse treatment effects. We also mention that live cells were counted using the LUNA-FL™ Dual Fluorescence Cell Counter before reimplantation into secondary NSG mice. Moreover, we have now mentioned in the Figure 6 legend that growth curves represent the mean tumor volume for 8 tumors per treatment group. Concerning SI Figure 6E, we have now performed additional in vivo experiments (see comments from Reviewer 1) that improve the readability of the Kaplan-Meier plot.

6. In the Discussion, it should be acknowledged that systemic chemical HRDness may increase side effects of PARP inhibition.

We have now added this sentence in the discussion as follow: “Although we might assume greater toxicity of therapeutic strategies combining chemical HRDness with PARPi in tumor cells with high genomic instability, the adverse-event profile will have to be carefully evaluated.”

Minor comments

1. The text contains some small errors, please check for missing words and characters.

We have now carefully proof-read our manuscript

Not sure if the title should state ‘the STAT3/FA axis’ instead of ‘STAT3/FA axis’ and if it should be ‘PARP inhibition’ or ‘the PARP inhibitor olaparib’, instead of ‘PARP inhibitor’.

We made the correction

In the Methods, the heading ‘Statistical’ should be ‘Statistics’ or something like ‘Statistical Analyses’.

We made the correction